# Discounted Beta–Bernoulli Reward Estimation
# for Sample-Efficient Reinforcement Learning with Verifiable Rewards

**Haechan Kim** [1 2]  **Soohyun Ryu** [1]  **Gyouk Chu** [1]  **Doohyuk Jang** [1]  **Eunho Yang** [1 3]

## Abstract

Reinforcement learning with verifiable rewards (RLVR) has emerged as an effective post-training paradigm for improving the reasoning capabilities of large language models. However, existing group-based RLVR methods often suffer from severe sample inefficiency. This inefficiency stems from reliance on point estimation of rewards from a small number of rollouts, leading to high estimation variance, variance collapse, and ineffective utilization of generated responses. In this work, we reformulate RLVR from a statistical estimation perspective by modeling rewards as samples drawn from a policy-induced distribution and casting advantage computation as the problem of estimating the reward distribution from finite data. Building on this view, we propose **D**iscounted **B**eta–**B**ernoulli (**DBB**) reward estimation, which leverages historical reward statistics for the non-stationary distribution. Although biased, the resulting estimator exhibits reduced and stable variance and theoretically avoids variance collapse. Under mild non-stationarity, it also achieves a lower mean squared error than standard point estimation, as we characterize analytically and verify empirically. Across six in-distribution and three out-of-distribution reasoning benchmarks, GRPO with DBB consistently outperforms naive GRPO and strong recent baselines, including the replay-based RePO and the variance-collapse-aware GRESO and DAPO. Relative to GRPO, it achieves average Acc@8 improvements of 3.43/2.32 points in-distribution and 10.05/8.34 points out-of-distribution on the 1.7B and 8B models, respectively, without additional computational cost or memory usage.

[1] Kim Jaechul Graduate School of AI, KAIST, Daejeon, South Korea [2] KRAFTON, Seoul, South Korea [3] AITRICS, Seoul, South Korea. Correspondence to: Eunho Yang <eunhoy@kaist.ac.kr>.

*Proceedings of the $43^{rd}$ International Conference on Machine Learning*, Seoul, South Korea. PMLR 306, 2026. Copyright 2026 by the author(s).

## 1. Introduction

RLVR (Lambert et al., 2025) has recently emerged as a key post-training paradigm for enhancing the complex reasoning capabilities of large language models (Plaat et al., 2025) and improving their performance on downstream tasks (Guo et al., 2025; Yang et al., 2025; Yu et al., 2025b). Since most RLVR algorithms, including Group Relative Policy Optimization (GRPO; Shao et al., 2024), rely on group-based advantage estimation (Yu et al., 2025a; Liu et al., 2025; Zheng et al., 2025a; Zhao et al., 2025), they require generating multiple responses per prompt, which can account for a large fraction of the overall training time, often approaching 50% (Le et al., 2025). Despite this substantial computational cost, many existing group-based RLVR algorithms fail to utilize information from the generated responses efficiently.

This sample inefficiency can be attributed to two fundamental characteristics of RLVR. The first characteristic is that the variance of rewards can collapse to zero when all generated responses receive identical rewards under group-relative estimation. This *variance collapse* issue not only wastes the computational cost of response generation but also eliminates meaningful training signals within the batch (Zhang et al., 2025). To address this issue, approaches such as dynamic sampling (Yu et al., 2025a) and GRPO with Efficient Selective Rollout (GRESO; Zheng et al., 2025b) have been proposed. However, dynamic sampling requires several times the rollout budget, and GRESO cannot fully resolve the problem since it probabilistically filters prompts that are likely to result in variance collapse.

The second characteristic is that, due to the on-policy nature of RLVR algorithms, information from all generated responses is discarded after a single gradient update. Recent replay-based methods (Li et al., 2025; Zhan et al., 2025) attempt to reuse rollouts generated from the training history. However, unbiased reuse of off-policy data requires importance sampling, which in turn necessitates storing all token-level probabilities under historical policies and performing additional forward passes on the current policy. As a result, replay-based approaches introduce substantial GPU memory overhead and additional computational cost, limiting their practical scalability.

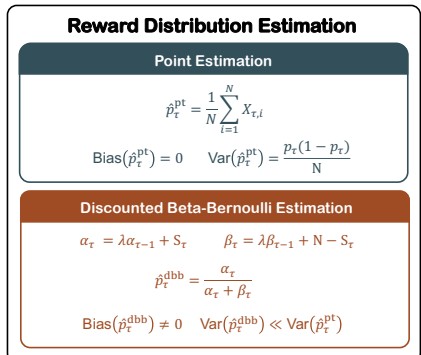
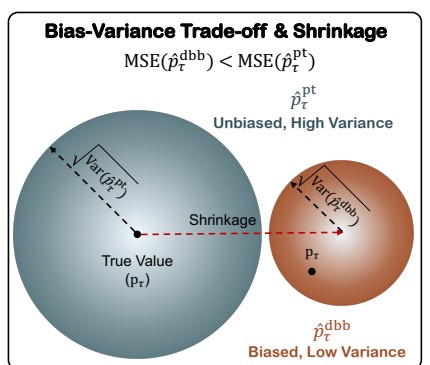
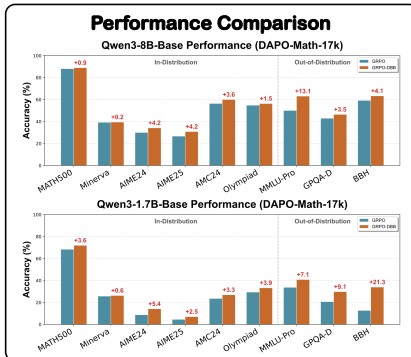

*Figure 1.* Comparison between point estimation and DBB estimation. By trading a small bias for substantial variance reduction via shrinkage, DBB estimation achieves lower mean squared error. Compared to naive GRPO using point estimation, GRPO with DBB estimation consistently demonstrates superior performance across all benchmarks and both model scales.

These two sources of sample inefficiency primarily arise because most group-based RLVR algorithms rely on point estimators that consider only the rewards from the current rollout group when estimating the underlying reward distribution for advantage estimation. While such estimators can be reliable when the number of sampled responses per group is sufficiently large (Casella & Berger, 2024), practical RLVR settings often operate in low-sample regimes due to computational constraints. In these settings, point estimators remain unbiased but suffer from high variance, which can lead to variance collapse and unstable training dynamics.

To address this limitation, we adopt a statistical perspective that models rewards as stochastic outcomes drawn from a distribution induced by the policy. Instead of relying on point estimation, we employ a Bayesian framework (Gelman et al., 2013) to model uncertainty explicitly and incorporate temporal reward dynamics. Within this framework, we propose the **D**iscounted **B**eta–**B**ernoulli (**DBB**) reward estimation, which tracks the evolving reward distribution by discounting historical observations. While DBB introduces a small bias, it significantly reduces variance and avoids variance collapse, thereby providing more stable and informative training signals (Figure 1). Empirically, DBB yields a lower mean squared error with respect to the true reward distribution than point estimation in low-sample scenarios, provided that the discount factor $\lambda$ is chosen in a reasonable range.

We evaluate GRPO with the DBB reward estimator (GRPO-DBB) on two model scales, Qwen3-1.7B-Base and Qwen3-8B-Base (Yang et al., 2025), across six in-distribution mathematical reasoning benchmarks, including MATH500, Minerva, AIME24/25, AMC24, and OlympiadBench. GRPO-DBB consistently outperforms naive GRPO and other baselines across all benchmarks and model sizes. Specifically, compared to GRPO, GRPO-DBB achieves average Acc@8 improvements of 3.43 and 2.32 points on Qwen3-1.7B-Base

and Qwen3-8B-Base, respectively. These improvements are statistically significant across nearly all benchmarks over four independent runs. Furthermore, on three out-of-distribution reasoning benchmarks such as MMLU-Pro, GPQA-Diamond, and Big-Bench Hard, GRPO-DBB yields average Acc@8 gains of 10.05 and 8.34 points over GRPO on 1.7B and 8B models, respectively. GRPO-DBB also outperforms recent strong baselines including the replay-based RePO and the variance-collapse-aware GRESO and DAPO.

## 2. Preliminaries

We briefly introduce RLVR and its commonly used baseline, GRPO, to establish the foundation for our method.

### 2.1. Reinforcement Learning with Verifiable Rewards

Given a prompt $q \sim \mathcal{D}$ from the training dataset and a policy $\pi_\theta$, a generated response is $o \sim \pi_\theta(\cdot \mid q)$. In RLVR, the reward function $r(\cdot)$ is typically defined as a binary signal indicating whether the response contains a correct answer.

$$r(o, q) = \begin{cases} 1, & \text{if } o \text{ contains the answer of } q \\ 0, & \text{otherwise.} \end{cases} \quad (1)$$

### 2.2. Group Relative Policy Optimization (GRPO)

GRPO (Shao et al., 2024) is a foundational baseline for RLVR that eliminates the need for an explicit value model and Generalized Advantage Estimation (Schulman et al., 2017). Instead, it relies on group-relative normalization to estimate advantages.

For a given prompt $q$, GRPO samples $N$ independent responses $\{o_i\}_{i=1}^N$ from the old policy $\pi_{\theta_{\text{old}}}$ and computes advantages by normalizing the corresponding rewards $\{r(o_i, q)\}_{i=1}^N$ within the group:

$$\hat{A}_i = \frac{r(o_i, q) - \mu}{\sigma}, \qquad (2)$$

where $\mu = \frac{\sum_{i=1}^{N} r(o_i, q)}{N}$ and $\sigma^2 = \frac{\sum_{i=1}^{N}(r(o_i,q)-\mu)^2}{N-1}$. The resulting advantage $\hat{A}_i$ is broadcast to all tokens in the response, i.e., $\hat{A}_{i,t} = \hat{A}_i$ for all token positions $t$.

The policy is then updated by maximizing a clipped surrogate objective based on estimated advantages:

$$\mathcal{J}_{\text{GRPO}}(\theta) = \mathbb{E}_{q \sim \mathcal{D}, \{o_i\}_{i=1}^{N} \sim \pi_{\theta_{\text{old}}}(\cdot|q)} \left[ \frac{1}{N} \sum_{i=1}^{N} \frac{1}{|o_i|} \sum_{t=1}^{|o_i|} \right.$$

$$\left. \min \left( w_{i,t}(\theta) \, \hat{A}_{i,t}, \text{clip}(w_{i,t}(\theta), 1 - \epsilon, 1 + \epsilon) \, \hat{A}_{i,t} \right) \right], \qquad (3)$$

where $\epsilon$ is the clipping hyperparameter and $w_{i,t}(\theta) = \frac{\pi_\theta(o_{i,t}|q,o_{i,<t})}{\pi_{\theta_{\text{old}}}(o_{i,t}|q,o_{i,<t})}$ denotes the per-token importance weight. Following DAPO (Yu et al., 2025a), we omit the KL divergence term between the online and reference policies, as it may restrict exploration.

## 3. Method

Many group-based RLVR methods (Shao et al., 2024; Liu et al., 2025; Xie et al., 2025; Le et al., 2025) primarily focus on the design of advantage estimators, while paying comparatively little attention to the more fundamental problem of reward estimation that underlies advantage computation. In practice, computational and memory constraints limit the number of sampled responses per prompt, making accurate reward estimation inherently difficult.

In contrast, we introduce a new perspective that reformulates RLVR through the lens of statistical estimation (Section 3.1). Specifically, we view the reward not as a deterministic signal, but as a distribution induced by the policy, and frame advantage computation as a problem of estimating this distribution from finite samples. Building on this reformulation, we propose **D**iscounted **B**eta–**B**ernoulli (**DBB**) reward estimation for RLVR, which estimates the non-stationary reward distribution by leveraging historical rewards (Section 3.2). Despite introducing bias, DBB reduces estimator variance and, under mild non-stationarity, achieves lower MSE than point estimation. Moreover, it fundamentally prevents variance collapse and preserves informative training signals (Section 3.3).

### 3.1. Reward Estimation as Distributional Inference

For a given prompt $q \sim \mathcal{D}$ and training step $\eta$ with corresponding epoch $\tau$, each response generated by the policy $\pi_{\theta_{\eta,\text{old}}}$ produces a binary reward indicating whether the response is correct. Under our formulation, this reward is

naturally modeled as a Bernoulli random variable

$$X_{\tau,i} \sim \text{Bernoulli}(p_\tau), \qquad (4)$$

where $p_\tau = \mathbb{P}(X_{\tau,i} = 1 \mid q, \pi_{\theta_{\eta,\text{old}}})$ denotes the probability of obtaining a correct response.

Within this framework, group-based RLVR algorithms such as GRPO and Dr.GRPO can be interpreted as implicitly estimating the reward distribution from a finite set of rollout samples. Given $N$ observed rewards $\{X_{\tau,i}\}_{i=1}^{N}$, existing approaches rely on *point estimation* of $p_\tau$ via the empirical mean

$$\hat{p}_\tau^{\text{pt}} = \frac{1}{N} \sum_{i=1}^{N} X_{\tau,i}. \qquad (5)$$

The corresponding reward variance is then estimated using the sample variance,

$$\widehat{\text{Var}}^{\text{pt}}(X_\tau) = \frac{\sum_{i=1}^{N} (X_{\tau,i} - \hat{p}_\tau^{\text{pt}})^2}{N-1} = \frac{N \cdot \hat{p}_\tau^{\text{pt}}(1 - \hat{p}_\tau^{\text{pt}})}{N-1}. \qquad (6)$$

Since the estimated variance $\widehat{\text{Var}}^{\text{pt}}$ is determined by the point estimator (mean) $\hat{p}_\tau^{\text{pt}}$, it suffices to consider only the point estimator in our analysis. Concretely, the point estimator $\hat{p}_\tau^{\text{pt}}$ has expectation and variance as follows:

$$\mathbb{E}[\hat{p}_\tau^{\text{pt}} \mid p_\tau] = p_\tau, \qquad (7)$$

$$\text{Var}(\hat{p}_\tau^{\text{pt}} \mid p_\tau) = \frac{p_\tau(1 - p_\tau)}{N}. \qquad (8)$$

While the point estimator is unbiased, its variance grows as the number of rollouts $N$ decreases. As a result, individual rollout outcomes can exert disproportionate influence on the estimate, causing the estimator to fluctuate widely. Moreover, the estimated variance can collapse to zero when all sampled rewards are identical. Since group-relative methods compute advantages directly from these reward estimates, such estimation noise naturally propagates to the advantage signal, leading to unstable policy updates.

### 3.2. Discounted Beta–Bernoulli Reward Estimation

To estimate the Bernoulli reward distribution beyond empirical averaging, we adopt a well-established Bayesian perspective that leverages the conjugacy between the Beta prior and the Bernoulli likelihood. In this setting, the reward probability is modeled using a Beta prior and updated via a Beta posterior after observing rollout outcomes, leading to the naive Beta–Bernoulli model defined below.

**Definition 1** (Beta–Bernoulli Reward Model). *To model uncertainty in the Bernoulli reward distribution, we place a Beta prior over the reward probability:*

$$p_\tau \sim \text{Beta}(\alpha_\tau, \beta_\tau), \qquad (9)$$

**Algorithm 1** Discounted Beta–Bernoulli Reward Estimation

**input** LLM policy $\pi_\theta$, training dataset $\mathcal{D}$

1: Initialize $(\alpha_0^q, \beta_0^q) \leftarrow (1,1) \quad \forall q \in \mathcal{D}$
2: **for** epoch $\tau = 1, \ldots, I$ **do**
3:     **for** iteration $= 1, \ldots, M$ **do**
4:         Sample a minibatch $\mathcal{D}_b \subset \mathcal{D}$
5:         Set old policy $\pi_{\theta_{\mathrm{old}}} \leftarrow \pi_\theta$
6:         **for** each prompt $q \in \mathcal{D}_b$ **do**
7:             Sample outputs $\{o_i\}_{i=1}^N \overset{\text{i.i.d.}}{\sim} \pi_{\theta_{\mathrm{old}}}(\cdot \mid q)$
8:             Compute rewards $\{X_i = r(o_i, q)\}_{i=1}^N$
9:             Update posterior parameters:

10:
$$\alpha_\tau^q \leftarrow \lambda \alpha_{\tau-1}^q + \sum_{i=1}^N X_i,$$
$$\beta_\tau^q \leftarrow \lambda \beta_{\tau-1}^q + N - \sum_{i=1}^N X_i$$

11:             Compute advantage $\hat{A}_i$ with $(\alpha_\tau^q, \beta_\tau^q)$ and $X_i$
12:         **end for**
13:         **for** update $= 1, \ldots, U$ **do**
14:             Update $\pi_\theta$ by maximizing the objective (Eq. 3)
15:         **end for**
16:     **end for**
17: **end for**

where $(\alpha_\tau, \beta_\tau)$ *denote the posterior parameters for a prompt at epoch $\tau$, with initialization $\alpha_0 = \beta_0 = 1$.*

*Given $N$ rollouts with $S_\tau$ successes, the posterior distribution is given by:*

$$\alpha_\tau = \alpha_{\tau-1} + S_\tau, \qquad \beta_\tau = \beta_{\tau-1} + N - S_\tau. \quad (10)$$

However, the above model assumes a stationary reward distribution. In RLVR, this assumption does not hold: due to the on-policy nature of training, the policy $\pi_\theta$ evolves over time, inducing a non-stationary reward distribution. As a result, historical observations may become outdated and should not be weighted equally with recent rollouts.

To address this challenge, we introduce the **D**iscounted **B**eta–**B**ernoulli (**DBB**) reward model, which gradually discounts past information to estimate the reward distribution accurately.

**Definition 2** (Discounted Beta–Bernoulli Reward Model). *Given posterior parameters $(\alpha_{\tau-1}, \beta_{\tau-1})$ and $N$ rollouts with $S_\tau$ successes at epoch $\tau$, the update is defined as*

$$\alpha_\tau = \lambda \alpha_{\tau-1} + S_\tau, \qquad \beta_\tau = \lambda \beta_{\tau-1} + N - S_\tau, \quad (11)$$

*where the discount factor $\lambda \in (0, 1]$ controls the influence of historical observations.*

Under the DBB reward model, we estimate the mean and

variance of the Bernoulli reward distribution as follows:

$$\hat{p}_\tau^{\mathrm{dbb}} = \frac{\alpha_\tau}{\alpha_\tau + \beta_\tau}, \quad (12)$$

$$\widehat{\mathrm{Var}}^{\mathrm{dbb}}(X_\tau \mid \alpha_\tau, \beta_\tau) = \frac{\alpha_\tau \beta_\tau}{(\alpha_\tau + \beta_\tau)^2}. \quad (13)$$

To understand the statistical behavior of the DBB estimator, we analyze its expectation and variance:

$$\mathbb{E}\big[\hat{p}_\tau^{\mathrm{dbb}} \mid p_\tau, \alpha_{\tau-1}, \beta_{\tau-1}\big] = w\,\mu_{\tau-1} + (1 - w)\,p_\tau, \quad (14)$$

$$\mathrm{Var}\big(\hat{p}_\tau^{\mathrm{dbb}} \mid p_\tau, \alpha_{\tau-1}, \beta_{\tau-1}\big) = (1 - w)^2 \frac{p_\tau(1 - p_\tau)}{N}, \quad (15)$$

where $\mu_{\tau-1} = \frac{\alpha_{\tau-1}}{\alpha_{\tau-1}+\beta_{\tau-1}}$ denotes the historical posterior mean and $w = \frac{\lambda(\alpha_{\tau-1}+\beta_{\tau-1})}{\lambda(\alpha_{\tau-1}+\beta_{\tau-1})+N}$ controls the contribution of past observations.

These show that the DBB estimator introduces bias through shrinkage toward the historical mean, but substantially reduces variance relative to the point estimator (Equation (8)). Importantly, unlike estimated variance based on small-sample rollouts, the posterior variance in DBB cannot collapse to zero, ensuring stable and informative reward signals throughout training. Algorithm 1 summarizes the overall training procedure of the group-based RLVR algorithm with DBB.

### 3.3. Mean Squared Error of the DBB Estimator

To assess the effectiveness of our DBB estimator in predicting the reward distribution, we compute its mean squared error (MSE). To make the estimator's dependence on the underlying reward probabilities $\{p_1, \ldots, p_\tau\}$ explicit, we re-express its expectation and variance, originally formulated in terms of the posterior parameters $(\alpha_{\tau-1}, \beta_{\tau-1})$.

Based on the update defined in Equation (11), posterior parameters can be expressed as

$$\alpha_\tau = \lambda^\tau \alpha_0 + \sum_{k=1}^\tau \lambda^{\tau-k} S_k, \;\; \beta_\tau = \lambda^\tau \beta_0 + \sum_{k=1}^\tau \lambda^{\tau-k}(N - S_k).$$

For convenience, define the sum of the posterior parameters as $H_\tau$:

$$H_\tau = \alpha_\tau + \beta_\tau = \lambda^\tau(\alpha_0 + \beta_0) + N \sum_{k=1}^\tau \lambda^{\tau-k}. \quad (16)$$

Under our formulation (Equation (4)), the number of successful rollouts at epoch $k$ satisfies $S_k \sim \mathrm{Binomial}(N, p_k)$. Given this, the expectation and variance of the DBB estimator can be written as follows:

$$\mathbb{E}\big[\hat{p}_\tau^{\mathrm{dbb}} \mid p_{1:\tau}\big] = \sum_{k=0}^\tau c_k p_k, \quad (17)$$

$$\mathrm{Var}\big(\hat{p}_\tau^{\mathrm{dbb}} \mid p_{1:\tau}\big) = \frac{\sum_{k=1}^{\tau} \lambda^{2(\tau-k)} N p_k(1-p_k)}{H_\tau^2}, \quad (18)$$

where $p_0 = \frac{\alpha_0}{\alpha_0+\beta_0}$ is the reward probability of the initial prior, and the weights $c_0 = \frac{\lambda^\tau(\alpha_0+\beta_0)}{H_\tau}$ and $c_k = \frac{N\lambda^{\tau-k}}{H_\tau}$ are constants. A detailed derivation is provided in Appendix B.

Combining Equations (17) and (18), the MSE of the DBB estimator under non-stationary rewards is

$$\mathrm{MSE}\big(\hat{p}_\tau^{\mathrm{dbb}} \mid p_{1:\tau}\big) = \big(\mathbb{E}\big[\hat{p}_\tau^{\mathrm{dbb}} \mid p_{1:\tau}\big] - p_\tau\big)^2 \\ + \mathrm{Var}\big(\hat{p}_\tau^{\mathrm{dbb}} \mid p_{1:\tau}\big). \quad (19)$$

In contrast, the MSE of the point estimator depends only on the current reward distribution

$$\mathrm{MSE}\big(\hat{p}_\tau^{\mathrm{pt}} \mid p_\tau\big) = \frac{p_\tau(1-p_\tau)}{N}, \quad (20)$$

and is therefore more prone to high variance, especially in low-sample settings.

The bias of the DBB estimator becomes smaller when the underlying reward probabilities evolve gradually, as past values $\{p_k\}_{k<\tau}$ remain close to the current $p_\tau$, while its variance is jointly controlled by the discount factor $\lambda$ and the number of rollouts $N$. Whether the DBB estimator achieves a lower MSE than the point estimator therefore depends on the relative magnitudes of bias and variance, which we empirically evaluate in Section 5.3.

## 4. Experiments

We empirically demonstrate that DBB estimation is effective in the RLVR setting. Section 4.1 describes the models, datasets, baselines, and detailed setup used in our experiments, while Section 4.2 presents the experimental results.

### 4.1. Experimental Settings

**Models & Datasets.** We conduct experiments on two model sizes, Qwen3-1.7B-Base and Qwen3-8B-Base (Yang et al., 2025), to investigate the effect of model scales. For both model scales, we use DAPO-Math-17k (Yu et al., 2025a) as the training dataset. As in-distribution evaluation benchmarks for mathematical reasoning, we consider six widely used datasets: MATH500 (Hendrycks et al., 2021), Minerva (Lewkowycz et al., 2022), AIME24/25, AMC24 (Li et al., 2024), and OlympiadBench (Olympiad; He et al., 2024). To further assess out-of-distribution generalization, we evaluate performance on MMLU-Pro (Wang et al., 2024), GPQA-Diamond (GPQA-D; Rein et al., 2023), and Big-Bench Hard (BBH; Suzgun et al., 2022).

**Baselines.** The DBB reward estimation aims to improve reward distribution estimation by partially leveraging historical information. To isolate the effect of DBB, we first compare GRPO with the DBB estimator (GRPO-DBB) and naive GRPO, which relies on point estimation. In addition, we include three representative baselines. RePO (Li et al., 2025) is a history-based method that replays full historical trajectories. GRESO (Zheng et al., 2025b) and DAPO (Yu et al., 2025a) are variance-collapse-aware methods: GRESO probabilistically filters prompts likely to lead to variance collapse, while DAPO uses dynamic sampling to discard zero-variance prompts. For all baselines, we follow the best configurations reported in their respective papers. To ensure a fair comparison, we standardize the total number of rollouts used throughout training across all methods.

**Training & Evaluation Setups.** All training experiments are conducted using the `verl`[1] framework, and `Math-Verify`[2] is employed to extract and verify final answers. Experiments with Qwen3-1.7B-Base and Qwen3-8B-Base are conducted on $4\times$H200 and $8\times$H200 GPUs, respectively, except for RePO. Due to the requirement for additional GPU memory, RePO is trained on $8\times$H200 GPUs for both model scales. The rollout batch size is set to 128, and the gradient update batch size is set to 64. We sample 8 responses per query from the on-policy model and train the model for a total of four epochs. For evaluation, we conduct four independent runs per method with different random seeds, sampling 8 responses per query in each run (i.e., 32 responses per prompt in total). We report Acc@8 as the evaluation metric. Each cell reports the mean and standard deviation across the four runs, and statistical significance is assessed via one-sided paired $t$-tests over per-prompt accuracies. The training setup details for each baseline and the sampling parameters used for validation and evaluation are also provided in Appendix C.

### 4.2. Main Results

Table 1 reports Acc@8 results across six mathematical reasoning in-distribution benchmarks and three general reasoning out-of-distribution (OOD) benchmarks for five different RLVR algorithms. For GRPO-DBB, the discount factor $\lambda$ is set to 0.5 for Qwen3-1.7B-Base and 0.75 for Qwen3-8B-Base; these values are empirically selected based on performance trends under varying $\lambda$. Further details are provided in Section 5.2.

For in-distribution evaluation, GRPO-DBB consistently outperforms all four baselines across all benchmarks and model scales. Compared to GRPO, Acc@8 improves by an average of 3.43 points for Qwen3-1.7B-Base and 2.32 points for Qwen3-8B-Base. These improvements are statistically significant in nearly every cell ($p < 0.05$). This indicates that,

---

[1] https://github.com/verl-project/verl
[2] https://github.com/huggingface/Math-Verify

*Table 1.* In-distribution (ID) and out-of-distribution (OOD) evaluation results trained on DAPO-Math-17k. For a fair comparison, all methods are trained with the same total number of rollouts. Evaluation is conducted over four independent runs with Acc@8 per run (i.e., 32 responses per prompt in total); each cell reports mean $\pm$ standard deviation across the four runs. $\Delta$ rows report the absolute Acc@8 improvement of GRPO-DBB over each baseline together with a one-sided paired $t$-test $p$-value ($H_1$: GRPO-DBB > baseline).

| Method | In-Distribution | | | | | | | Out-of-Distribution | | | |
| | MATH500 | Minerva | AIME24 | AIME25 | AMC24 | Olympiad | Avg. | MMLU-Pro | GPQA-D | BBH | Avg. |
|---|---|---|---|---|---|---|---|---|---|---|---|
| | **Qwen3-8B-Base** trained with **DAPO-Math-17k** | | | | | | | | | | |
| GRPO | 87.87±0.36 | 39.34±0.29 | 29.17±0.68 | 25.31±2.16 | 57.71±1.44 | 54.54±0.19 | 48.99±0.46 | 50.16±0.33 | 45.27±1.69 | 59.18±0.33 | 51.54±0.55 |
| RePO | 86.22±0.31 | 35.02±0.47 | 27.29±0.80 | 23.44±1.04 | 56.25±0.70 | 52.70±0.44 | 46.82±0.07 | 53.29±0.55 | 46.34±1.74 | 42.44±0.30 | 47.35±0.80 |
| GRESO | 87.01±0.20 | 33.88±0.60 | 27.36±0.79 | 22.78±1.09 | 57.31±0.57 | 52.81±0.17 | 46.86±0.12 | 62.81±0.39 | 48.53±0.43 | 65.17±0.63 | 58.83±0.40 |
| DAPO | 86.00±0.07 | 37.08±0.30 | 31.06±1.19 | 21.02±1.14 | 52.67±0.92 | 51.64±0.14 | 46.58±0.16 | 59.28±0.67 | 45.05±1.79 | 63.41±0.29 | 55.91±0.46 |
| GRPO-DBB | **88.71**±0.18 | **39.46**±0.51 | **33.60**±2.05 | **29.02**±2.53 | **60.54**±1.00 | **56.53**±0.25 | **51.31**±0.53 | **63.12**±0.38 | **49.43**±2.11 | **67.09**±0.13 | **59.88**±0.66 |
| $\Delta$ w.r.t. GRPO | +0.84 (0.0073) | +0.12 (0.2015) | +4.44 (0.0114) | +3.71 (0.0063) | +2.83 (0.0018) | +1.99 (0.0009) | +2.32 (0.0007) | +12.96 (0.0000) | +4.17 (0.0053) | +7.91 (0.0000) | +8.34 (0.0000) |
| $\Delta$ w.r.t. RePO | +2.49 (0.0008) | +4.44 (0.0000) | +6.31 (0.0072) | +5.58 (0.0234) | +4.29 (0.0038) | +3.83 (0.0001) | +4.49 (0.0002) | +9.84 (0.0000) | +3.09 (0.0024) | +24.65 (0.0000) | +12.53 (0.0000) |
| $\Delta$ w.r.t. GRESO | +1.70 (0.0011) | +5.57 (0.0006) | +6.24 (0.0077) | +6.24 (0.0052) | +3.23 (0.0102) | +3.72 (0.0002) | +4.45 (0.0002) | +0.32 (0.0032) | +0.90 (0.1806) | +1.92 (0.0045) | +1.05 (0.0077) |
| $\Delta$ w.r.t. DAPO | +2.71 (0.0000) | +2.37 (0.0011) | +2.54 (0.0220) | +8.00 (0.0010) | +7.87 (0.0012) | +4.89 (0.0001) | +4.73 (0.0001) | +3.84 (0.0013) | +4.39 (0.0019) | +3.68 (0.0001) | +3.97 (0.0001) |
| | **Qwen3-1.7B-Base** trained with **DAPO-Math-17k** | | | | | | | | | | |
| GRPO | 67.50±0.79 | 25.29±0.31 | 9.06±0.63 | 4.48±0.21 | 23.40±1.39 | 29.57±0.37 | 26.55±0.26 | 33.62±0.09 | 21.90±1.04 | 12.71±0.17 | 22.74±0.32 |
| RePO | 67.27±0.55 | 23.78±0.12 | 7.60±1.50 | 5.94±0.71 | 22.57±1.39 | 29.00±0.43 | 26.03±0.29 | 36.58±0.73 | 25.25±0.99 | **30.37**±0.41 | 30.73±0.32 |
| GRESO | 67.60±0.05 | 23.76±0.20 | 8.19±1.37 | 6.39±0.79 | 23.61±1.96 | 30.91±0.15 | 26.74±0.09 | 35.68±0.35 | 27.15±1.24 | 26.67±0.18 | 29.83±0.36 |
| DAPO | 66.69±0.13 | 24.34±0.23 | 9.72±0.86 | 5.42±0.90 | 19.72±0.45 | 28.70±0.06 | 25.77±0.33 | 33.71±0.69 | 24.49±0.86 | 21.05±0.12 | 26.42±0.42 |
| GRPO-DBB | **71.67**±0.22 | **26.43**±0.67 | **13.75**±1.13 | **7.40**±1.04 | **27.31**±1.90 | **33.36**±0.50 | **29.98**±0.28 | **40.81**±0.11 | **27.89**±1.46 | 29.67±0.89 | **32.79**±0.79 |
| $\Delta$ w.r.t. GRPO | +4.17 (0.0004) | +1.14 (0.0419) | +4.69 (0.0063) | +2.92 (0.0037) | +3.90 (0.0158) | +3.79 (0.0007) | +3.43 (0.0001) | +7.19 (0.0000) | +5.98 (0.0085) | +16.97 (0.0000) | +10.05 (0.0002) |
| $\Delta$ w.r.t. RePO | +4.40 (0.0002) | +2.64 (0.0015) | +6.14 (0.0007) | +1.46 (0.0219) | +4.74 (0.0202) | +4.35 (0.0004) | +3.95 (0.0004) | +4.24 (0.0110) | +2.64 (0.0110) | −0.70 (0.8273) | +2.06 (0.0007) |
| $\Delta$ w.r.t. GRESO | +4.07 (0.0000) | +2.67 (0.0039) | +5.56 (0.0001) | +1.01 (0.0184) | +3.69 (0.0209) | +2.44 (0.0010) | +3.24 (0.0001) | +5.13 (0.0001) | +0.74 (0.0354) | +3.01 (0.0038) | +2.96 (0.0001) |
| $\Delta$ w.r.t. DAPO | +4.98 (0.0000) | +2.08 (0.0031) | +4.03 (0.0119) | +1.98 (0.0387) | +7.58 (0.0018) | +4.66 (0.0001) | +4.22 (0.0002) | +7.11 (0.0002) | +3.39 (0.0119) | +8.62 (0.0001) | +6.38 (0.0004) |

in the RLVR setting, replacing GRPO's point estimator with the DBB estimator is more effective for in-distribution downstream tasks. When compared to the other three baselines (RePO, GRESO, and DAPO), GRPO-DBB still achieves average in-distribution Acc@8 gains of 4.45–4.73 (8B) and 3.24–4.22 (1.7B) points. The three baselines take markedly different approaches to address sample inefficiency. RePO replays diverse forms of historical information, including tokens, token probabilities, and rewards. GRESO and DAPO instead modify the sampling/filtering strategy: GRESO probabilistically filters prompts that are likely to lead to variance collapse, and DAPO dynamically resamples within the current iteration to discard zero-variance prompts. GRPO-DBB, in contrast, only changes the reward estimator by aggregating historical *reward* statistics, without replaying trajectories or modifying the sampling strategy. Despite this minimal departure from GRPO, its strong downstream performance suggests that the DBB estimator utilizes historical information both efficiently and effectively.

For out-of-distribution evaluation, GRPO-DBB consistently outperforms the baselines across nearly all OOD benchmarks and model scales. In particular, it achieves substantial average gains over GRPO of 10.05 and 8.34 points for Qwen3-1.7B-Base and Qwen3-8B-Base, respectively. Compared to the other three baselines (RePO, GRESO, and DAPO), GRPO-DBB still achieves average OOD Acc@8 gains of 1.05–12.53 (8B) and 2.06–6.38 (1.7B) points. With only two exceptions—BBH against RePO on the 1.7B model and GPQA-D against GRESO on the 8B model, where GRPO-DBB and the corresponding baseline are statistically indistinguishable—GRPO-DBB is statistically

*Table 2.* Effect of the discount factor $\lambda$ of GRPO-DBB on in-distribution Acc@8 performance. Qwen3-8B-Base achieves the best performance at $\lambda = 0.75$, while Qwen3-1.7B-Base performs best at $\lambda = 0.5$.

| Method | MATH500 | Minerva | AIME24 | AIME25 | AMC24 | Olympiad | Avg. |
|---|---|---|---|---|---|---|---|
| **Qwen3-8B-Base** trained with **DAPO-Math-17k** | | | | | | | |
| $\lambda = 1.0$ | 89.08 | 37.91 | 33.33 | 26.25 | 59.44 | 55.19 | 50.20 |
| $\lambda = 0.75$ | 88.71 | 39.46 | 33.60 | 29.02 | 60.54 | 56.53 | **51.31** |
| $\lambda = 0.5$ | 89.35 | 38.56 | 33.33 | 27.08 | 59.17 | 56.19 | 50.61 |
| $\lambda = 0.25$ | 88.82 | 39.98 | 30.83 | 25.83 | 60.83 | 56.34 | 50.44 |
| GRPO | 87.87 | 39.34 | 29.17 | 25.31 | 57.71 | 54.54 | 48.99 |
| **Qwen3-1.7B-Base** trained with **DAPO-Math-17k** | | | | | | | |
| $\lambda = 1.0$ | 67.30 | 25.55 | 9.58 | 3.33 | 23.33 | 28.65 | 26.29 |
| $\lambda = 0.75$ | 72.28 | 25.60 | 11.25 | 7.50 | 26.39 | 32.68 | 29.28 |
| $\lambda = 0.5$ | 71.67 | 26.43 | 13.75 | 7.40 | 27.31 | 33.36 | **29.98** |
| $\lambda = 0.25$ | 72.22 | 25.97 | 14.17 | 6.25 | 22.50 | 32.88 | 29.00 |
| GRPO | 67.50 | 25.29 | 9.06 | 4.48 | 23.40 | 29.57 | 26.55 |

significantly superior to every baseline across all OOD benchmark–model-scale combinations ($p < 0.05$). These results demonstrate that the DBB estimation provides a strong advantage in improving generalization, which can be attributed to the Bayesian approach's ability to account for uncertainty.

## 5. Analysis & Discussion

In this section, we provide an in-depth analysis of the training behavior and estimation properties of GRPO with the DBB reward estimator. We first examine training dynamics to understand how DBB influences optimization stability, exploration behavior, and reward progression over time (Section 5.1). We then conduct ablation studies on the discount factor $\lambda$ to clarify its role in adapting to different learning

regimes (Section 5.2). Next, we empirically analyze the mean squared error (MSE) of DBB relative to point estimation, connecting estimation accuracy to downstream performance (Section 5.3). Finally, we demonstrate that DBB can be readily integrated into alternative advantage formulations, highlighting its generality beyond GRPO (Section 5.4).

## 5.1. Training Dynamics

Figure 2 illustrates the validation accuracy and training dynamics of GRPO-DBB and GRPO during training. For GRPO-DBB, the discount factor $\lambda$ is set to $0.5$ for Qwen3-1.7B-Base and $0.75$ for Qwen3-8B-Base. For both model scales, GRPO-DBB achieves higher validation accuracy and training rewards than GRPO throughout most of the training process.

Examining the response length and entropy over the course of training, we observe that GRPO-DBB consistently produces longer responses than GRPO, while the entropy under GRPO grows rapidly, indicating increasingly unstable exploration behavior. We attribute this pattern to how DBB handles zero-variance prompts. For such prompts, the group-relative advantage of GRPO collapses to zero and yields no training signal. In contrast, DBB still produces a non-zero advantage: for all-incorrect groups it assigns a negative advantage that suppresses the failed trajectories, while for all-correct groups it assigns a positive advantage that reinforces the successful ones. By both suppressing unproductive trajectories and reinforcing successful ones on zero-variance prompts, DBB concentrates probability mass in the policy's token tree, lowering entropy, and shifts exploration from the breadth-wise mode (diversifying trajectories) toward the depth-wise mode (extending them), naturally yielding the longer responses observed. We interpret this mechanism as a key driver of the gains of GRPO-DBB. Importantly, the lower entropy does not compromise sample diversity at evaluation time: as shown in Appendix F, GRPO-DBB matches or exceeds GRPO in Best@8 across both model scales, indicating that meaningful exploratory paths are preserved even as exploration is sharpened.

## 5.2. Ablation Study on $\lambda$

The hyperparameter $\lambda$ controls the extent to which the estimation of the Bernoulli parameter $p_\tau$ for a given query at epoch $\tau$ depends on historical rollout statistics, namely $\alpha_{\tau-1}$ and $\beta_{\tau-1}$. When the learning dynamics are fast and the discrepancy between $p_\tau$ and $p_{\tau-1}$ is large, a smaller $\lambda$ is generally more effective, as it places greater emphasis on recent observations and reduces estimation bias. Conversely, when the learning dynamics are slower and more stationary, a larger $\lambda$ tends to yield better performance by incorporating more historical information and thereby reducing estimation variance.

To examine this effect empirically, we train GRPO-DBB with $\lambda = 1.0, 0.75, 0.5,$ and $0.25$ for both model scales and report the in-distribution Acc@8 in Table 2. For Qwen3-8B-Base, $\lambda = 0.75$ attains the best average Acc@8 of $51.31$, whereas for Qwen3-1.7B-Base the best is at $\lambda = 0.5$ with $29.98$, corresponding to gains of $2.32$ and $3.43$ points over naive GRPO, respectively. Every $\lambda$ setting outperforms naive GRPO except $\lambda = 1.0$ on Qwen3-1.7B-Base, confirming that incorporating historical reward statistics through DBB is broadly beneficial. Notably, the optimal $\lambda$ shifts with model scale, with the smaller model favoring a smaller $\lambda$.

This scale dependence aligns with the training reward dynamics shown in Figure 2. For Qwen3-1.7B-Base, the training reward exhibits sustained changes throughout training, indicating continuously evolving estimates of $p_\tau$; in contrast, for Qwen3-8B-Base, the training reward rises rapidly within approximately the first $0.3$ epoch and remains relatively stationary thereafter. The faster-evolving dynamics on the smaller model thus favor a smaller $\lambda$ to reduce bias, while the more stationary dynamics on the larger model favor a larger $\lambda$ to reduce variance, consistent with the bias–variance trade-off above. More broadly, weaker models or more challenging datasets tend to incur larger between-epoch shifts in the reward distribution and therefore benefit from a smaller $\lambda$, whereas once the model fits the training distribution well, the reward distribution becomes nearly stationary and bounded, allowing a larger $\lambda$ to further reduce MSE. To corroborate this trend, we additionally conduct a $\lambda$ search on Qwen3-0.6B-Base in Appendix G. There, $\lambda = 0.25$ attains the best performance, consistent with the rule of thumb that *smaller models tend to favor smaller $\lambda$*. Based on this observation, we cautiously suggest a starting point for larger models. For 30B+ models trained on datasets of similar difficulty to DAPO-Math-17k, $\lambda$ close to 1 may be a reasonable starting point, while harder datasets are likely to benefit from a smaller $\lambda$.

## 5.3. Analysis of MSE

To evaluate how accurately the point estimator $\hat{p}_\tau^{\text{pt}}$ and the DBB estimator $\hat{p}_\tau^{\text{dbb}}$ approximate the Bernoulli parameter $p_\tau$ during training, we conduct an empirical estimation error analysis. Since obtaining the exact ground-truth value of $p_\tau$ would require infinitely many samples, we instead approximate it using the empirical mean of rewards from a finite number of rollouts.

During GRPO training of Qwen3-1.7B-Base, when a randomly selected 10% subset of training prompts $\mathcal{Q}_\tau$ appears in a rollout batch, we additionally sample 128 responses for each such prompt using the policy at that training step. These additional samples are used exclusively for analysis and are not incorporated into training. We then define the

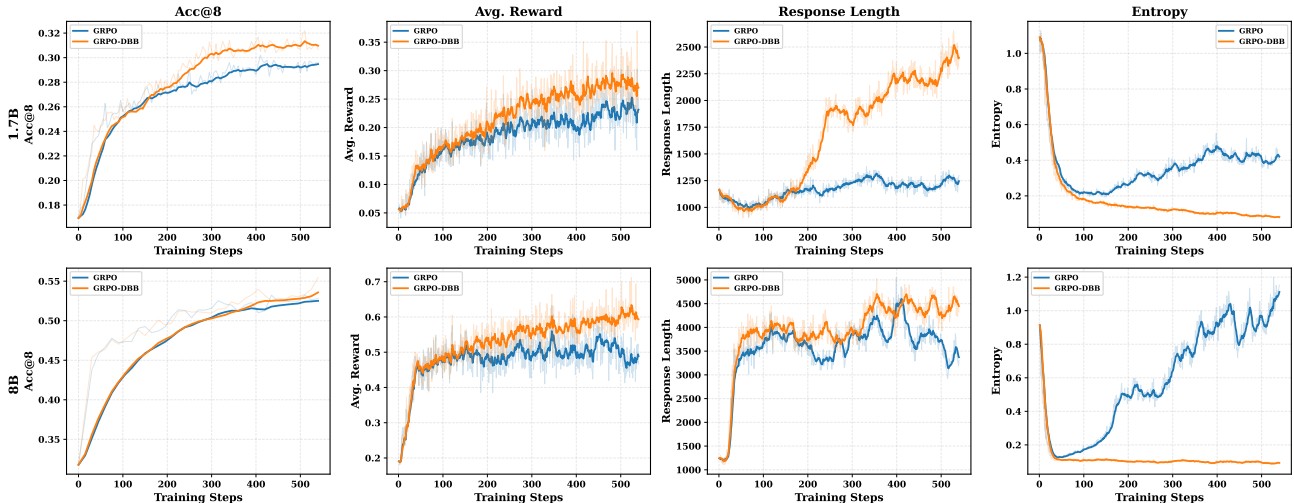

*Figure 2.* Training dynamics of naive GRPO and GRPO with the DBB estimator (GRPO-DBB) on Qwen3-1.7B-Base (top) and Qwen3-8B-Base (bottom). GRPO-DBB achieves higher validation Acc@8 and training rewards, while maintaining longer responses with controlled entropy compared to GRPO, indicating more stable exploration during training.

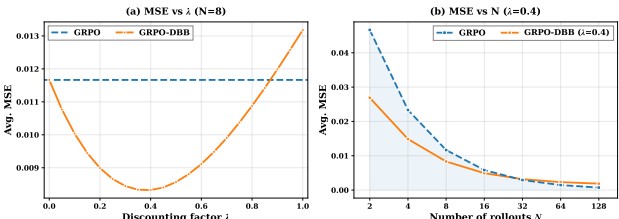

*Figure 3.* MSE as a function of the discount factor $\lambda$ and the number of rollouts $N$. The DBB estimator yields lower MSE than the point estimator across a wide range of $\lambda$, and it achieves lower MSE for rollout budgets up to $N = 16$ when $\lambda = 0.4$.

empirical mean of the resulting rewards as the reference value $\tilde{p}_\tau(q)$.

$$\tilde{p}_\tau(q) \triangleq \frac{1}{128} \sum_{i=1}^{128} X_{\tau,i}, \tag{21}$$

where $o_{\tau,i} \sim \pi_{\theta_\tau}(\cdot \mid q)$ and $X_{\tau,i} = r(o_{\tau,i}, q)$. Since training is conducted for a total of four epochs, this procedure yields a sequence of reference values $\{\tilde{p}_\tau(q)\}_{\tau=1}^4$ for each query.

Using $\tilde{p}_\tau(q)$ as a surrogate for the true Bernoulli parameter, we measure the average MSE of the DBB estimator and the point estimator following Equations (19) and (20), respectively.

**Effect of $\lambda$ on MSE.** Since the value of the DBB estimator depends on the discount factor $\lambda$, we empirically analyze how the MSE varies as a function of $\lambda$. As shown in Figure 3(a), the MSE attains its minimum at $\lambda = 0.4$, and for most values of $\lambda$—except for $\lambda = 0.95$ and $\lambda = 1.0$—the

DBB estimator yields a lower average MSE than the point estimator.

This trend is closely aligned with downstream performance. As reported in Table 2, GRPO-DBB achieves the best performance at $\lambda = 0.5$, while at $\lambda = 1.0$ it performs slightly worse than GRPO with the point estimator. These observations mirror the behavior of the MSE across different values of $\lambda$. Taken together, the results indicate that more accurate reward distribution estimation is strongly correlated with improved RLVR performance.

**Effect of $N$ on MSE.** We further examine how the MSE varies as the number of rollouts $N$ increases. Figure 3(b) reports the average MSE as a function of $N$ when $\lambda = 0.4$. The results show that the DBB estimator consistently outperforms the point estimator for $N$ up to 16. This suggests that, in regimes where computational resource constraints necessitate small rollout budgets, the DBB estimator can serve as an effective alternative to point estimation.

**Comparison with Alternative Shrinkage Estimators.** We further benchmark DBB against three alternative shrinkage estimators: per-prompt exponential moving average (EMA), Laplace smoothing, and an online empirical-Bayes (EB) variant that shares a Beta prior across prompts. Appendix H shows that DBB attains the lowest minimum MSE and improves on the point estimator over a broader $\lambda$ range than either EMA or Laplace smoothing. Appendix I further shows that EB, while avoiding $\lambda$ tuning, still attains a higher MSE than the minimum MSE of the DBB estimator. In other words, since the DBB estimator empirically achieves the lowest MSE, this indicates that it is a superior estimator compared to the three alternatives.

*Table 3.* In-distribution evaluation results on Qwen3-1.7B-Base trained with DAPO-Math-17k. We apply the proposed discounted Beta–Bernoulli reward distribution estimation to the advantage term of Dr.GRPO and observe consistent performance improvements. $\Delta$ denotes the absolute Acc@8 improvement over naive Dr.GRPO.

| Method | MATH500 | Minerva | AIME24 | AIME25 | AMC24 | Olympiad | Avg. |
|---|---|---|---|---|---|---|---|
| **Qwen3-1.7B-Base** trained with **DAPO-Math-17k** | | | | | | | |
| Dr.GRPO | 69.00 | 25.14 | 9.17 | 7.92 | 23.89 | 32.78 | 27.98 |
| Dr.GRPO-DBB | **70.60** | **26.15** | **12.08** | **9.17** | **28.06** | **33.59** | **29.94** |
| $\Delta$ w.r.t. Dr.GRPO | **+1.60** | **+1.01** | **+2.92** | **+1.25** | **+4.17** | **+0.81** | **+1.96** |

### 5.4. Evaluation on Alternative Advantage Formulations

To examine whether the DBB estimation can be applied to other advantage formulations, we conduct experiments using the advantage term of Dr.GRPO, defined as $A_i = X_i - \mathbb{E}[\hat{p}_\tau]$. All experiments are conducted on Qwen3-1.7B-Base with the discount factor set to $\lambda = 0.5$ for the Dr.GRPO with the DBB estimation (Dr.GRPO-DBB). As shown in Table 3, Dr.GRPO-DBB consistently improves performance over the naive Dr.GRPO across all in-distribution benchmarks, yielding an average Acc@8 gain of 1.96 points. These results suggest that the DBB estimation is not limited to the advantage term of GRPO, but can be broadly applied to other RLVR algorithms that rely on point estimation, yielding effective and consistent performance gains.

## 6. Related Work

**Reinforcement Learning with Verifiable Rewards.** Reinforcement learning with verifiable rewards (RLVR) has emerged as an effective post-training paradigm for improving the reasoning capabilities of large language models by leveraging automatically verifiable reward signals (Lambert et al., 2025; Plaat et al., 2025). In particular, group-based RLVR algorithms, including GRPO and its variants (Shao et al., 2024; Yu et al., 2025a; Liu et al., 2025; Zheng et al., 2025a; Zhao et al., 2025), have demonstrated strong performance on reasoning benchmarks without relying on explicit value models (Yu et al., 2025b).

**Variance Collapse in Group-Based RLVR.** A well-known limitation of group-based RLVR methods is variance collapse, where the estimated reward variance becomes zero when all sampled responses receive identical rewards, eliminating the training signal and leading to ineffective policy updates (Yu et al., 2025a; Zheng et al., 2025b; Le et al., 2025; Zhang et al., 2025). Prior approaches such as Dynamic sAmpling Policy Optimization (DAPO; Yu et al., 2025a) and GRESO (Zheng et al., 2025b) mitigate this issue by modifying the rollout or sampling strategy. However, these methods either incur substantial additional computational cost or fail to fundamentally eliminate variance collapse due to their probabilistic or heuristic nature.

In contrast, our work addresses variance collapse at the reward estimator level by explicitly modeling uncertainty in the reward distribution. By maintaining a Beta posterior with strictly positive parameters, the DBB process theoretically prevents variance collapse without additional rollouts or changes to the sampling strategy.

**Replay- and History-Based RLVR Methods.** Replay-based methods aim to improve sample efficiency in RLVR by reusing historical rollouts, as exemplified by RePO (Li et al., 2025) and ExGRPO (Zhan et al., 2025). While potentially effective, these approaches require storing tokens and their probabilities under historical policies and performing additional forward passes, introducing non-trivial memory and computational overhead.

Our approach differs in that it leverages only historical reward statistics rather than full trajectory information, eliminating the need for additional GPU memory or extra forward passes. Empirical results demonstrate that reward signals alone can effectively capture useful historical information for improving RLVR performance.

## 7. Conclusion

We revisited RLVR from the perspective of reward distribution estimation and identified point estimation under limited rollouts as a key source of sample inefficiency and variance collapse. To this end, we proposed Discounted Beta–Bernoulli (DBB) reward estimation for RLVR, which leverages historical reward information for accurate estimation. Although biased, the DBB estimator achieves substantially lower variance and, under mild non-stationarity of rewards, a lower mean squared error than point estimation; it also theoretically avoids variance collapse and preserves informative training signals without additional rollouts or replay. Experiments across two model scales demonstrate consistent improvements over baselines on all in-distribution and out-of-distribution benchmarks. Our findings highlight reward distribution estimation as a critical yet underexplored component of effective RLVR and suggest that principled estimator design offers a promising direction for improving large-scale reinforcement learning of language models. As future work, we plan to design a dynamic discount factor $\lambda$ that adapts across different stages of training. We also view DBB as one instance of a broader discounted Bayesian reward-estimation framework. A natural next step is to pair the discounted update with other likelihoods (e.g., a discounted Gaussian) to handle continuous or dense process rewards.

## Acknowledgements

This work was supported by the Institute of Information & Communications Technology Planning & Evaluation (IITP) grant funded by the Korea government (MSIT) (RS-2019-II190075, Artificial Intelligence Graduate School Program(KAIST)) and National Research Foundation of Korea (NRF) grant funded by the Korea government (MSIT) (RS-2023-00209060, A Study on Optimization and Network Interpretation Method for LargeScale Machine Learning).

## Impact Statement

This paper presents work whose goal is to advance the field of machine learning. There are many potential societal consequences of our work, none of which we feel must be specifically highlighted here.

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

# A. Comparison of Statistics Between Point Estimation and Discounted Beta–Bernoulli Estimation

We present Table 4 that summarizes the statistics of point estimation and Discounted Beta–Bernoulli estimation, as discussed in Sections 3.1 and 3.2, in a concise and accessible manner.

*Table 4.* Comparison between point estimation and DBB estimation for reward distribution modeling. We denote the Bernoulli distribution by $\mathrm{Bern}(\cdot)$.

| | Point Estimation | DBB Estimation |
|---|---|---|
| $X_{\tau,i}$ | $X_{\tau,i} \sim \mathrm{Bern}(p_\tau)$ | $X_{\tau,i} \sim \mathrm{Bern}(p_\tau)$ |
| $p_\tau$ | fixed but unknown | $p_\tau \sim \mathrm{Beta}(\alpha_\tau, \beta_\tau)$ |
| $\hat{p}_\tau$ | $\hat{p}_\tau^{\mathrm{pt}} = \dfrac{1}{N}\sum_{i=1}^{N} X_{\tau,i}$ | $\hat{p}_\tau^{\mathrm{dbb}} = \dfrac{\alpha_\tau}{\alpha_\tau + \beta_\tau}$ |
| $\widehat{\mathrm{Var}}(X_\tau)$ | $\dfrac{N}{N-1}\,\hat{p}_\tau^{\mathrm{pt}}\big(1-\hat{p}_\tau^{\mathrm{pt}}\big)$ | $\hat{p}_\tau^{\mathrm{dbb}}\big(1-\hat{p}_\tau^{\mathrm{dbb}}\big)$ |
| $\mathbb{E}[\hat{p}_\tau]$ | $p_\tau$ | $w\mu_{\tau-1} + (1-w)p_\tau$ |
| $\mathrm{Bias}(\hat{p}_\tau)$ | $0$ | $w(\mu_{\tau-1} - p_\tau)$ |
| $\mathrm{Var}(\hat{p}_\tau)$ | $\dfrac{p_\tau(1-p_\tau)}{N}$ | $(1-w)^2\dfrac{p_\tau(1-p_\tau)}{N}$ |

# B. Derivation of the Mean, Variance, and MSE of the DBB Estimator

In this section, we provide a detailed derivation of Equations (17) and (18) in the main text. To avoid ambiguity caused by implicit conditioning, we re-express the analysis in terms of the conditional distribution given the sequence of true reward probabilities $\{p_1, p_2, \ldots, p_\tau\}$.

## B.1. Expansion of Posterior Parameters under Historical Rewards

At each training step $k$, rollout rewards satisfy

$$S_k \mid p_k \sim \mathrm{Binomial}(N, p_k), \qquad X_{k,i} \mid p_k \sim \mathrm{Bernoulli}(p_k), \tag{22}$$

and the DBB updates are given by

$$\alpha_k = \lambda\alpha_{k-1} + S_k, \qquad \beta_k = \lambda\beta_{k-1} + (N - S_k). \tag{23}$$

Unrolling the recursion yields

$$\alpha_\tau = \lambda^\tau \alpha_0 + \sum_{k=1}^{\tau} \lambda^{\tau-k} S_k, \tag{24}$$

and the total mass

$$H_\tau \triangleq \alpha_\tau + \beta_\tau = \lambda^\tau(\alpha_0 + \beta_0) + N\sum_{k=1}^{\tau} \lambda^{\tau-k}. \tag{25}$$

Crucially, when $N$ and $\lambda$ are fixed, $H_\tau$ is deterministic and does not depend on the random variables $\{S_k\}_{k=1}^{\tau}$. All stochasticity in the estimator arises from $\alpha_\tau$ alone.

## B.2. Mean of the DBB Estimator

The DBB estimator is defined as

$$\hat{p}_\tau^{\text{dbb}} = \frac{\alpha_\tau}{H_\tau}. \tag{26}$$

Since $H_\tau$ is deterministic,

$$\mathbb{E}\big[\hat{p}_\tau^{\text{dbb}} \mid p_{1:\tau}\big] = \frac{\mathbb{E}[\alpha_\tau \mid p_{1:\tau}]}{H_\tau}. \tag{27}$$

Using $\mathbb{E}[S_k \mid p_k] = N p_k$, we obtain

$$\mathbb{E}[\alpha_\tau \mid p_{1:\tau}] = \lambda^\tau \alpha_0 + N \sum_{k=1}^{\tau} \lambda^{\tau-k} p_k. \tag{28}$$

Therefore,

$$\mathbb{E}\big[\hat{p}_\tau^{\text{dbb}} \mid p_{1:\tau}\big] = \frac{\lambda^\tau \alpha_0 + N \sum_{k=1}^{\tau} \lambda^{\tau-k} p_k}{\lambda^\tau (\alpha_0 + \beta_0) + N \sum_{k=1}^{\tau} \lambda^{\tau-k}} \tag{29}$$

For interpretability, define weights

$$c_0 \triangleq \frac{\lambda^\tau (\alpha_0 + \beta_0)}{H_\tau}, \qquad c_k \triangleq \frac{N \lambda^{\tau-k}}{H_\tau}, \quad k = 1, \dots, \tau, \qquad p_0 \triangleq \frac{\alpha_0}{\alpha_0 + \beta_0}. \tag{30}$$

Then the estimator mean can be written as

$$\mathbb{E}\big[\hat{p}_\tau^{\text{dbb}} \mid p_{1:\tau}\big] = \sum_{k=0}^{\tau} c_k p_k, \qquad \sum_{k=0}^{\tau} c_k = 1, \tag{31}$$

revealing an exponentially weighted average of historical reward probabilities.

## B.3. Variance of the DBB Estimator

Again using the determinism of $H_\tau$,

$$\text{Var}\big(\hat{p}_\tau^{\text{dbb}} \mid p_{1:\tau}\big) = \frac{\text{Var}(\alpha_\tau \mid p_{1:\tau})}{H_\tau^2}. \tag{32}$$

Since $\{S_k\}$ are independent conditioned on $\{p_k\}$ and $\text{Var}(S_k \mid p_k) = N p_k (1 - p_k)$,

$$\text{Var}(\alpha_\tau \mid p_{1:\tau}) = \sum_{k=1}^{\tau} \lambda^{2(\tau-k)} N p_k (1 - p_k). \tag{33}$$

Hence,

$$\text{Var}\big(\hat{p}_\tau^{\text{dbb}} \mid p_{1:\tau}\big) = \frac{\sum_{k=1}^{\tau} \lambda^{2(\tau-k)} N p_k (1 - p_k)}{\big(\lambda^\tau (\alpha_0 + \beta_0) + N \sum_{k=1}^{\tau} \lambda^{\tau-k}\big)^2} \tag{34}$$

## B.4. Bias and Mean Squared Error of the DBB Estimator

By definition,

$$\text{MSE}\big(\hat{p}_\tau^{\text{dbb}} \mid p_{1:\tau}\big) = \text{Bias}^2 + \text{Var}. \tag{35}$$

Using Eq. (31), the conditional bias is

$$\text{Bias}\big(\hat{p}_\tau^{\text{dbb}} \mid p_{1:\tau}\big) = \sum_{k=0}^{\tau-1} c_k (p_k - p_\tau), \tag{36}$$

where the $k = \tau$ term vanishes since $p_\tau - p_\tau = 0$.

Combining Eqs. (34) and (36), we obtain

$$\mathrm{MSE}\big(\hat{p}_\tau^{\mathrm{dbb}} \mid p_{1:\tau}\big) = \left(\sum_{k=0}^{\tau-1} c_k(p_k - p_\tau)\right)^2 + \frac{\sum_{k=1}^{\tau} \lambda^{2(\tau-k)} N p_k(1 - p_k)}{H_\tau^2}, \tag{37}$$

where $H_\tau = \lambda^\tau(\alpha_0 + \beta_0) + N\sum_{k=1}^{\tau} \lambda^{\tau-k}$.

# C. Implementation Details

*Table 5.* Common training hyperparameter settings for all experiments.

| Hyperparameter | Qwen3-1.7B-Base | Qwen3-8B-Base |
|---|---|---|
| **Training Configuration** | | |
| Training batch size | 128 | 128 |
| Mini-batch size | 64 | 64 |
| Number of epochs | 4 | 4 |
| Total gradient steps | 1080 | 1080 |
| Samples per prompt | 8 | 8 |
| Max response length | 4096 | 8192 |
| **Sampling Configuration** | | |
| Training temperature | 1.0 | 1.0 |
| Training top-$p$ | 1.0 | 1.0 |
| Validation temperature | 0.6 | 0.6 |
| Validation top-$p$ | 0.95 | 0.95 |
| **Optimization** | | |
| Optimizer | AdamW | AdamW |
| Learning rate | $1 \times 10^{-6}$ | $1 \times 10^{-6}$ |
| LR warmup steps | 0 | 0 |
| LR scheduler | constant | constant |

In this section, we describe the experimental setup details for GRPO-DBB (and more generally any RLVR algorithm that applies the DBB process) and all baseline methods. All training experiments are conducted using the `verl` framework, and `Math-Verify` is employed to extract and normalize final answers from model responses. Training Qwen3-1.7B-Base is performed using 4×H200 GPUs, while Qwen3-8B-Base is trained using 8×H200 GPUs. Due to its substantially higher GPU memory requirements, RePO is trained using 8×H200 GPUs for both model scales.

All training experiments are conducted using the DAPO-Math-17k dataset. For training stability, we filter prompts to a maximum length of 1024 tokens. In addition, to facilitate analysis, we further restrict the dataset so that its size is an integer multiple of the rollout batch size, which is 128. As a result, out of the original 17,391 prompts, 17,280 are used for training.

The hyperparameters shared across all training experiments are summarized in Table 5. Our experimental setup follows configurations that are most commonly adopted in recent RLVR studies. Following DAPO, we remove the KL regularization term during training. Throughout this paper we use PPO clipping in its asymmetric clip-higher form, where the surrogate ratio $w_{i,t}(\theta)$ is clipped to $[1 - \epsilon_{\mathrm{low}}, 1 + \epsilon_{\mathrm{high}}]$. For all baselines (GRPO, RePO, GRESO, DAPO, and Dr.GRPO), we use $(\epsilon_{\mathrm{low}}, \epsilon_{\mathrm{high}}) = (0.2, 0.28)$, following DAPO. For GRPO-DBB and Dr.GRPO-DBB, we instead use a wider $(\epsilon_{\mathrm{low}}, \epsilon_{\mathrm{high}}) = (0.98, 0.98)$ to avoid over-clipping the larger advantages produced by the DBB estimator. A detailed analysis showing that this choice does not by itself explain the performance gains is provided in Appendix D.

For GRPO-DBB and Dr.GRPO-DBB, we initialize the Beta posterior with the symmetric uninformative prior $(\alpha_0, \beta_0) = (1, 1)$, which avoids injecting bias toward either outcome while keeping the prior mass small relative to the on-policy batch. A detailed MSE sensitivity analysis over $(\alpha_0, \beta_0)$ and the corresponding $\lambda$ is provided in Appendix E.

For GRESO, we follow the original implementation and use the reported best probabilistic-filtering thresholds; for DAPO we use the dynamic-sampling configuration recommended in the original paper, retaining the same total number of rollouts as the other methods. For RePO, which introduces additional hyperparameters and configuration options, we follow the

optimal settings recommended in the original paper. Specifically, we set the number of replay samples to 8, use a replay cache size of 16, and adopt the reward-oriented replay strategy.

For both evaluation and validation, we use the sampling parameters specified in Table 5. To reduce training overhead, the validation set is restricted to AIME24/25, AMC24, Minerva, and MATH500. Evaluation on OlympiadBench and the out-of-distribution benchmarks (MMLU-Pro, GPQA-Diamond, and Big-Bench Hard) is performed only on the best checkpoint selected for each experiment. For statistical significance, we additionally perform four independent evaluation runs per method by varying the evaluation seed; each cell in Table 1 reports the mean and standard deviation across the four runs, and $p$-values are obtained from one-sided paired $t$-tests over per-prompt accuracies.

## D. Isolating the Effect of the Clipping Range

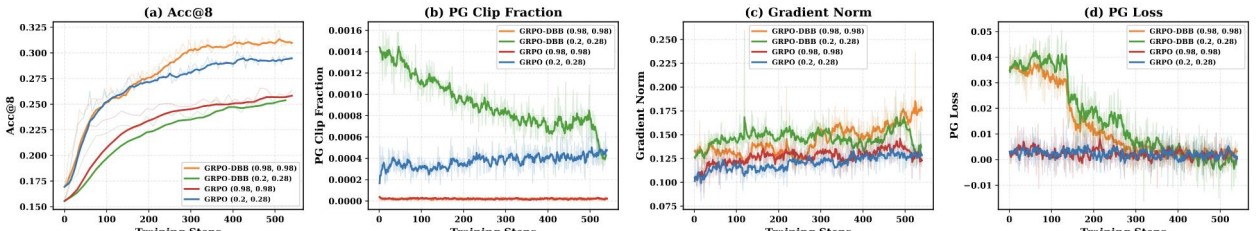

*Figure 4.* Training dynamics of Qwen3-1.7B-Base with GRPO and GRPO-DBB under clipping ranges $(0.2, 0.28)$ and $(0.98, 0.98)$. From left to right: (a) validation Acc@8, (b) policy-gradient clip fraction, (c) gradient norm, and (d) policy-gradient loss. GRPO-DBB suffers excessive clipping under the standard range; once the range is widened, it recovers strong performance. GRPO under the wider range shows no improvement, confirming that the gains of GRPO-DBB are due to the DBB estimator rather than the clip range.

We further investigate whether the performance gains of GRPO-DBB stem from the DBB estimator itself or from the wider clipping range $(0.98, 0.98)$ used for it. To this end, we train GRPO and GRPO-DBB on Qwen3-1.7B-Base under both $(\epsilon_{\text{low}}, \epsilon_{\text{high}}) = (0.2, 0.28)$ and $(0.98, 0.98)$. Figure 4 compares four diagnostics across the four resulting runs: (a) validation Acc@8, (b) policy-gradient clip fraction, (c) gradient norm, and (d) policy-gradient loss.

We first explain why GRPO-DBB requires a wider clip range to perform well. The DBB estimator removes GRPO's zero-sum advantage normalization. As a result, the advantages produced by DBB have a larger dynamic range than those of GRPO, except in the degenerate zero-variance case. Larger advantages scale the same per-token gradients by a larger factor, leading to larger importance ratios. Figure 4(c) and (d) confirm this empirically: GRPO-DBB exhibits larger gradient norms and larger policy-gradient losses than GRPO regardless of the clip range. These effects are therefore *intrinsic to the estimator*, not artifacts of the clip range.

A direct consequence is that, under the standard $(0.2, 0.28)$ range, GRPO-DBB suffers a 2–3× higher clip fraction than GRPO (Figure 4(b)). This suppresses a substantial portion of meaningful training signal and noticeably degrades Acc@8 (Figure 4(a)). Once the clip range is widened to $(0.98, 0.98)$, GRPO-DBB recovers its expected gains and outperforms GRPO, which already shows that the improvements are not merely a consequence of taking larger gradient steps. Critically, GRPO under the same wider $(0.98, 0.98)$ range shows similar or even smaller gradient norms and no improvement in policy gradient loss compared to GRPO under $(0.2, 0.28)$. This rules out the hypothesis that a wider clip range mechanically improves performance. Accordingly, in all main-table experiments we report each method with its own best clipping configuration.

## E. Sensitivity to the Prior Hyperparameters $(\alpha_0, \beta_0)$

We additionally measure the MSE of the DBB estimator with respect to the prior hyperparameters $(\alpha_0, \beta_0)$ and the discount factor $\lambda$ under the same setup as Section 5.3.

Figure 5(a) shows a heatmap of $\min_\lambda \text{MSE}(\hat{p}_\tau^{\text{dbb}} \mid p_{1:\tau}; \alpha_0, \beta_0)$ over $(\alpha_0, \beta_0)$. For most reasonable prior choices, the achievable MSE remains close to the optimum. Performance degrades only when $\alpha_0 + \beta_0$ is large, where a strong prior diminishes the influence of the $N$ on-policy rewards. In that regime, the estimator becomes more sensitive to $\lambda$, which can amplify bias and inflate MSE (Figure 5(b)).

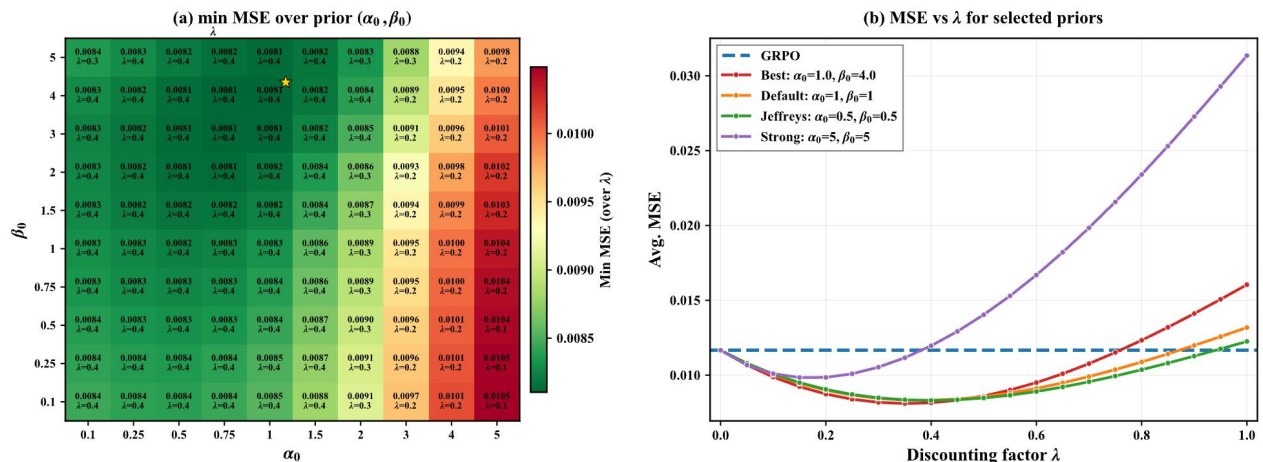

*Figure 5.* (a) Heatmap of $\min_\lambda$ MSE of the DBB estimator over $(\alpha_0, \beta_0)$. (b) MSE versus $\lambda$ for selected priors. The default symmetric prior $(\alpha_0, \beta_0) = (1, 1)$ is competitive with the best prior and robust over a wide range of $\lambda$.

In practice we initialize the Beta posterior with the symmetric uninformative prior $(\alpha_0, \beta_0) = (1, 1)$. This choice is both reasonable and robust. In the absence of prior knowledge about $p_\tau$, symmetric initialization avoids injecting bias toward either outcome. At the same time, it keeps the prior mass small relative to typical batch sizes. As Figure 5(b) shows, $(1, 1)$ achieves a relatively low MSE across a wide range of $\lambda$ and a competitive minimum. Overall, MSE is sensitive to $(\alpha_0, \beta_0)$ and to the corresponding $\lambda$, but this sensitivity is not a practical concern as long as pathological prior choices are avoided.

## F. Mean@8 and Best@8 Validation Curves

To further characterize the exploration behavior of GRPO-DBB, we plot both Mean@8 (= Acc@8) and Best@8 on the validation set throughout training for both model scales in Figure 6. For Qwen3-1.7B-Base, GRPO-DBB attains higher Best@8 than GRPO for most of training. For Qwen3-8B-Base, GRPO temporarily achieves a higher Best@8 during the middle stage but is eventually overtaken by GRPO-DBB in the final stage. These results indicate that the lower entropy of GRPO-DBB (Figure 2) does not compromise but in fact improves Best@8 in the long run.

To quantify this, we additionally evaluate the best checkpoint of each method on the six in-distribution benchmarks with four independent evaluation seeds (i.e., 32 responses per prompt). GRPO attains Best@8 scores of $61.75 \pm 0.92$ (8B) and $39.79 \pm 0.26$ (1.7B), while GRPO-DBB attains $63.61 \pm 1.16$ (8B) and $43.73 \pm 0.43$ (1.7B). The Best@8 improvements of GRPO-DBB are well outside the standard deviations, so the gains are statistically meaningful. This further indicates that the sharper exploration of GRPO-DBB preserves the diversity of useful response trajectories.

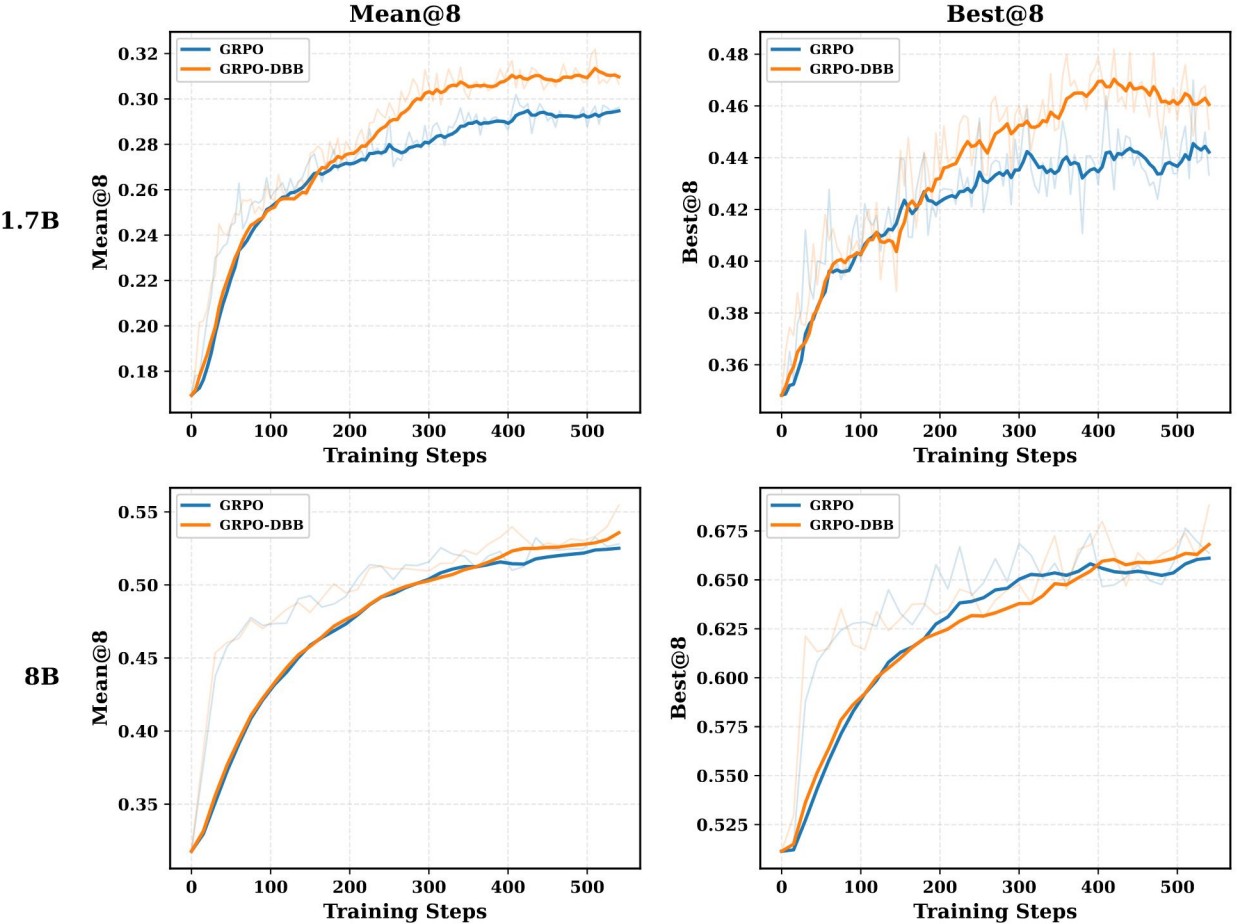

*Figure 6.* Validation performance of Qwen3-1.7B-Base (top) and Qwen3-8B-Base (bottom) trained with GRPO and GRPO-DBB. The left column shows Mean@8 (= Acc@8) and the right column shows Best@8 on the validation set. Despite producing lower entropy during training, GRPO-DBB matches or exceeds GRPO in Best@8 across both model scales.

## G. Discount-Factor Search on Qwen3-0.6B-Base

To further investigate how the optimal discount factor $\lambda$ depends on model scale, we conduct an additional $\lambda$ search on a smaller backbone, Qwen3-0.6B-Base, trained with DAPO-Math-17k. Table 6 reports Acc@8 on the six in-distribution benchmarks for $\lambda \in \{0.25, 0.5, 0.75, 1.0\}$.

*Table 6.* Effect of the discount factor $\lambda$ of GRPO-DBB on in-distribution Acc@8 performance for Qwen3-0.6B-Base trained on DAPO-Math-17k. The best average performance is attained at $\lambda = 0.25$.

| Method | MATH500 | Minerva | AIME24 | AIME25 | AMC24 | Avg. |
|---|---|---|---|---|---|---|
| **Qwen3-0.6B-Base** trained with **DAPO-Math-17k** | | | | | | |
| $\lambda = 1.0$ | 47.12 | 14.25 | 2.92 | 0.00 | 10.83 | 15.02 |
| $\lambda = 0.75$ | 48.05 | 14.52 | 2.08 | 0.42 | 10.28 | 15.07 |
| $\lambda = 0.5$ | 47.95 | 14.11 | 2.08 | 1.25 | 11.94 | 15.47 |
| $\lambda = 0.25$ | 49.98 | 14.15 | 3.33 | 1.67 | 8.61 | **15.55** |

The optimal $\lambda$ for the 0.6B model is $0.25$, smaller than the optimal values $\lambda^\star = 0.5$ for 1.7B and $\lambda^\star = 0.75$ for 8B. Together with our analysis in Section 5.2, this supports a simple rule of thumb: *smaller (or weaker) backbones tend to favor smaller $\lambda$.* The reason is that their reward distribution shifts more rapidly during training, so the bias induced by a larger $\lambda$ outweighs the variance reduction it provides. We extrapolate this rule cautiously to larger models. For 30B+ models trained on datasets of comparable difficulty to DAPO-Math-17k, $\lambda$ close to 1 is a reasonable starting point. Harder datasets (where shifts are larger) are likely to benefit from a smaller $\lambda$.

## H. Comparison with Per-Prompt EMA and Laplace Smoothing

A natural question is whether simpler shrinkage estimators could achieve similar gains as the DBB estimator. To address this, we additionally derive and evaluate the per-prompt exponential moving average (EMA) and Laplace smoothing (Lap) estimators of the success probability $p_\tau$. For a direct comparison against the DBB estimator on a comparable axis, we parameterize the shrinkage strength of each estimator by a single hyperparameter $\lambda$.

**Per-Prompt EMA Estimator.** We define the per-prompt EMA recursively as

$$\hat{p}_\tau^{\text{EMA}} = (1 - \lambda)\frac{S_\tau}{N} + \lambda\hat{p}_{\tau-1}^{\text{EMA}}, \qquad \hat{p}_1^{\text{EMA}} = \frac{S_1}{N}. \tag{38}$$

Unrolling the recursion gives $\hat{p}_\tau^{\text{EMA}} = \sum_{k=1}^\tau w_k (S_k/N)$ with weights

$$w_1 = \lambda^{\tau-1}, \qquad w_k = (1 - \lambda)\lambda^{\tau-k} \ (k \geq 2), \qquad \sum_{k=1}^\tau w_k = 1. \tag{39}$$

Conditional on $p_{1:\tau}$ and using $S_k \sim \text{Binomial}(N, p_k)$, its conditional MSE is

$$\text{MSE}(\hat{p}_\tau^{\text{EMA}} \mid p_{1:\tau}) = \left(\sum_{k=1}^{\tau-1} w_k (p_k - p_\tau)\right)^2 + \sum_{k=1}^\tau w_k^2 \frac{p_k(1 - p_k)}{N}. \tag{40}$$

**Laplace Smoothing.** We define the Laplace smoothing as

$$\hat{p}_\tau^{\text{Lap}} = \frac{S_\tau + \lambda}{N + 2\lambda}, \tag{41}$$

whose conditional MSE is

$$\text{MSE}(\hat{p}_\tau^{\text{Lap}} \mid p_\tau) = \frac{\lambda^2 + (N - 4\lambda^2)p_\tau(1 - p_\tau)}{(N + 2\lambda)^2}. \tag{42}$$

**Empirical Comparison.** Following the same protocol of Section 5.3, we sweep $\lambda$ for each estimator. As shown in Figure 7, the DBB estimator achieves the lowest MSE of $0.0083$ at $\lambda = 0.4$, outperforming all other methods. EMA attains its best MSE of $0.0086$ at $\lambda = 0.3$, which is $2.67\%$ higher than that of the DBB estimator; however, the MSE of EMA rises rapidly as $\lambda$ grows, and the range of $\lambda$ over which EMA outperforms the point estimator is relatively narrow. Similarly, Laplace smoothing attains its best MSE of $0.0108$ at $\lambda = 0.3$, $21.71\%$ higher than that of the DBB estimator, and likewise outperforms the point estimator only within a narrow $\lambda$ range. Because the DBB estimator not only achieves the lowest minimum MSE, but also maintains performance improvements over the point estimator across a broader range of $\lambda$, it is overall the most favorable option among these light-weight shrinkage estimators.

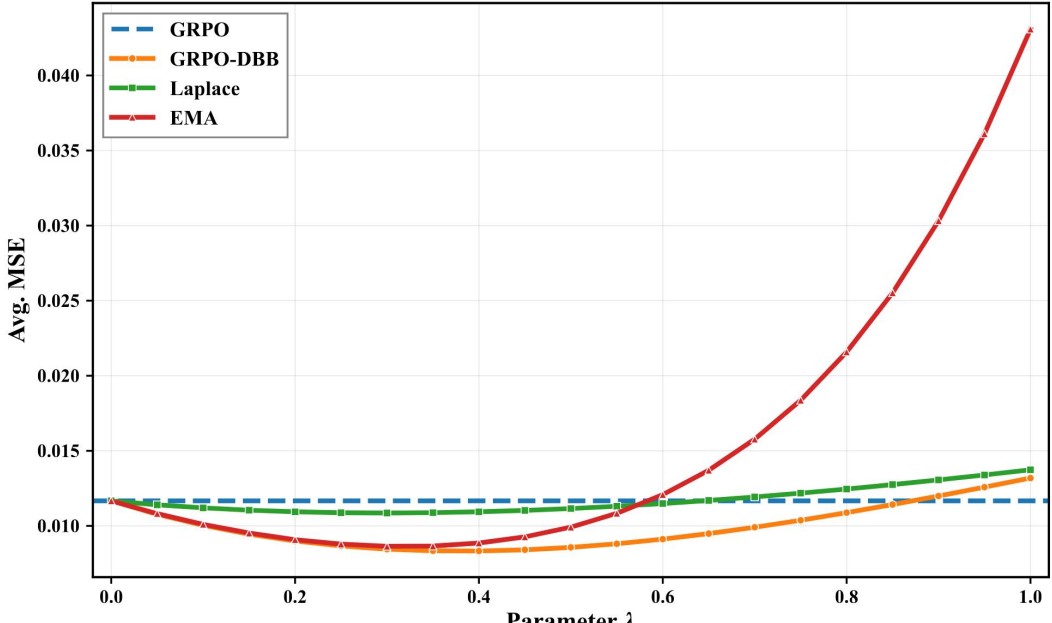

*Figure 7.* MSE versus $\lambda$ for per-prompt exponential moving average (EMA), Laplace smoothing (Laplace), point estimator (GRPO), and DBB estimator (GRPO-DBB). The DBB estimator attains the lowest minimum MSE at $\lambda = 0.4$ and improves on the point estimator over a broader range of $\lambda$ than EMA or Laplace.

## I. Comparison with Hierarchical / Empirical-Bayes Prior

We also consider Empirical Bayes (EB) as a representative approach for introducing a hierarchical / shared prior across prompts. The hyperparameters of the shared Beta prior, $\alpha_{\mathrm{EB}}$ and $\beta_{\mathrm{EB}}$, are estimated online via method-of-moments, yielding $\hat{\alpha}_{\mathrm{EB}}(s)$ and $\hat{\beta}_{\mathrm{EB}}(s)$ at online step $s$. For a given prompt with $N$ on-policy rewards summing to $S_\tau$ (so that the point estimate is $\hat{p}_\tau^{\mathrm{pt}} = S_\tau/N$), the EB estimator can be written as

$$\hat{p}_\tau^{\mathrm{EB}} = \frac{N\,\hat{p}_\tau^{\mathrm{pt}} + \hat{\alpha}_{\mathrm{EB}}(s)}{N + \hat{\alpha}_{\mathrm{EB}}(s) + \hat{\beta}_{\mathrm{EB}}(s)}. \tag{43}$$

Let

$$\nu_{\mathrm{EB}}(s) = \hat{\alpha}_{\mathrm{EB}}(s) + \hat{\beta}_{\mathrm{EB}}(s), \qquad m_{\mathrm{EB}}(s) = \hat{\alpha}_{\mathrm{EB}}(s)/\nu_{\mathrm{EB}}(s), \tag{44}$$

denote the effective prior strength and the shared prior mean, respectively. Treating $(\hat{\alpha}_{\mathrm{EB}}(s), \hat{\beta}_{\mathrm{EB}}(s))$ as deterministic at step $s$ and using $S_\tau \sim \mathrm{Binomial}(N, p_\tau)$, the conditional MSE of the EB estimator can be expressed as

$$\mathrm{MSE}\big(\hat{p}_\tau^{\mathrm{EB}} \mid p_\tau\big) = \frac{\nu_{\mathrm{EB}}^2\,(m_{\mathrm{EB}} - p_\tau)^2 + N\,p_\tau(1 - p_\tau)}{(N + \nu_{\mathrm{EB}})^2}. \tag{45}$$

EB has the advantage of not requiring manual hyperparameter tuning, since the shared prior is estimated directly from the population of per-prompt posteriors. However, its minimum achievable MSE is higher than that of the DBB estimator

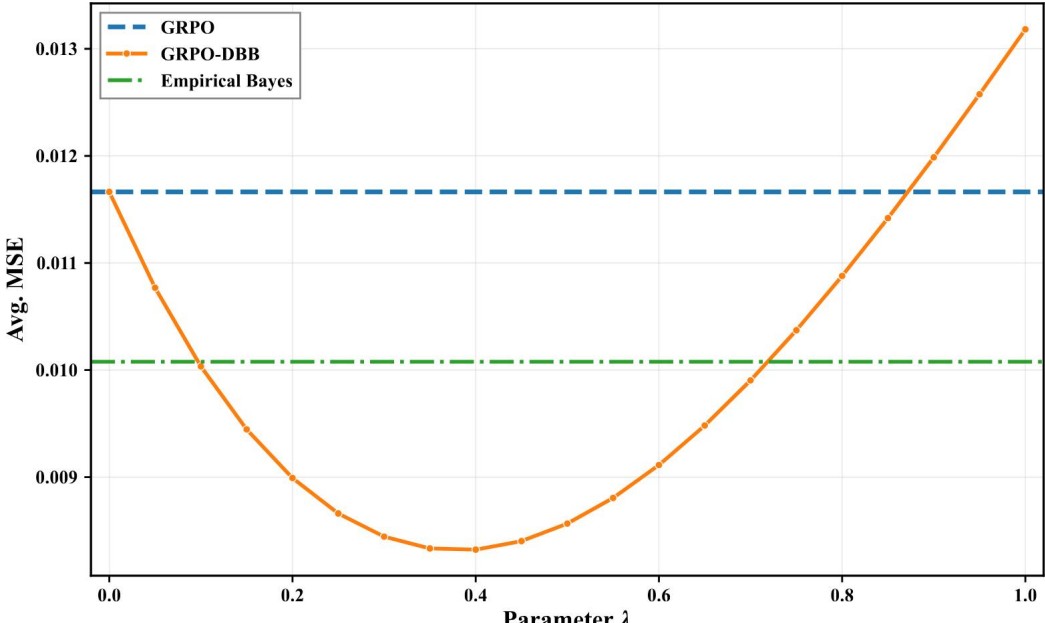

*Figure 8.* MSE versus $\lambda$ for the Empirical Bayes estimator with online method-of-moments (EB), point estimator (GRPO), and DBB estimator (GRPO-DBB). EB improves over the point estimator without per-prompt tuning, but DBB attains a lower minimum MSE.

(Figure 8). This suggests a trade-off between robustness (i.e., reduced dependence on $\lambda$) and optimal performance. Combining a hierarchical prior with the discounted update of the DBB estimator is a natural direction for future work.

