# OpenReview forum: "Discounted Beta–Bernoulli Reward Estimation for Sample-Efficient Reinforcement Learning with Verifiable Rewards"
_ICML.cc/2026/Conference — ICML 2026 regular_

### Official Review · Reviewer_pjp8 · 2026-03-09

**Soundness:** 3
**Presentation:** 3
**Significance:** 2
**Originality:** 2
**Overall Recommendation:** 3
**Confidence:** 4

**Summary:**

This paper focuses on the critical problem of sample inefficiency and variance collapse in group-based Reinforcement Learning with Verifiable Rewards (RLVR). The paper investigates a pertinent challenge. Moving away from standard point estimation, the authors reformulate advantage computation as a problem of estimating the reward distribution from finite data using a Bayesian framework. Specifically, they introduce the Discounted Beta-Bernoulli (DBB) reward estimation method, which leverages historical reward statistics while progressively discounting them to account for the non-stationary policy distribution. By integrating this estimator into the GRPO framework, the proposed approach reduces estimation variance, theoretically prevents variance collapse, and achieves a lower mean squared error. Extensive experiments on Qwen3-1.7B and 8B models show that GRPO with DBB consistently outperforms naive GRPO across six in-distribution and three out-of-distribution reasoning benchmarks, without incurring additional computational or memory overhead.

**Compliance With Llm Reviewing Policy:**

Affirmed.

**Final Justification:**

I appreciate the authors’ rebuttal, which addressed my main concerns. As a result, I maintain my original score and overall assessment.

**Key Questions For Authors:**

See weaknesses.

**Limitations:**

yes

**Strengths And Weaknesses:**

Strengths:

(1) The paper abandons the traditional research focus on the advantage estimator design of RLVR, and reinterprets RLVR from the perspective of statistical reward distribution estimation, which fills the research gap of neglecting the underlying reward estimation in existing group-based RLVR methods.

(2) The paper conducts a complete theoretical derivation of the expectation, variance, and MSE of the DBB estimator, and compares the statistical characteristics of DBB and point estimation in detail; it also theoretically proves that DBB can avoid variance collapse, which provides a solid theoretical foundation for the proposed method.

(3) The DBB method only uses historical reward statistical information, without additional GPU memory overhead and computational cost compared with replay-based methods such as RePO, which is more suitable for large-scale LLM training and has strong industrial application potential.

(4) The empirical evaluation is robust, showing substantial Acc@8 improvements of 12.49 and 6.92 points over naive GRPO on the 1.7B and 8B models respectively for challenging out-of-distribution benchmarks like MMLU-Pro and GPQA-Diamond.

Weaknesses:

(1) The paper only compares with GRPO and RePO, and does not conduct comparative experiments with other latest RLVR methods (e.g., ExGRPO, GRESO, DAPO), which makes it difficult to fully reflect the performance of DBB in the current RLVR method system.

(2) The empirical validation is currently limited to the Qwen3-1.7B-Base and Qwen3-8B-Base models. Since RLVR is increasingly applied to much larger frontier models, adding a brief discussion on whether the DBB mechanism and its hyperparameter sensitivities are expected to hold at scales of 30B+ parameters would broaden the paper's impact.

(3) While the current Beta-Bernoulli formulation elegantly handles binary verifiable rewards, the paper would benefit from a brief discussion on how this Bayesian perspective might eventually be extended to continuous or dense process rewards, which are becoming increasingly prevalent in the field.

(4) The performance metrics reported in the results show absolute Acc@8 improvements, but they do not include standard deviations or confidence intervals across multiple random seeds. Given the inherent stochasticity in reinforcement learning, adding statistical significance tests would significantly strengthen the reliability of the claims.

---

> ### Author Rebuttal · Authors · 2026-03-31
>
> We thank the reviewer for their time and constructive feedback. We respond to their comments below.
> Figures and tables are given in **[LINK](https://github.com/anonyauthor1872/2026-icml-rebuttal)**.
>
> **[W1+W4]**
> We conduct experiments on the baselines reported in the paper, GRESO and DAPO (i.e., GRPO with dynamic sampling), for both 1.7B and 8B models. For a fair comparison, we standardize the total number of rollouts used throughout training across all methods. The training configurations for each baseline are set according to the best (default) settings reported in their respective papers.
>
> For evaluation, we perform a total of four independent runs to assess statistical significance with different random seeds. (The tables reported in the paper are the results of a single run. ) In each run, Acc@8 is computed by sampling eight responses per prompt. In Table 6, the performance of each method is reported as the mean and standard deviation over the four runs. For the $\Delta$ values, we additionally report p-values obtained via a one-sided paired $t$-test. As a result, GRPO-DBB achieves statistically significant improvements over nearly all baselines across most benchmarks.
>
> **[W2]** While we attempted to evaluate the effectiveness of our method on the larger-scale model you suggested (30B+ parameters), we faced resource constraints regarding time and cost. We also personally find it unfortunate that we are unable to conduct such experiments.
>
> However, we attempt to provide additional insights by analyzing the relationship between model scale and the optimal choice of $\lambda$ through further experiments. For details, please refer to the response to the reviewer TVSP's W2. Based on this analysis, we cautiously suggest that for models at the 30B+ scale trained on datasets of similar difficulty to DAPO-17k, it may be beneficial to set $\lambda$ close to 1. For more challenging datasets, a smaller $\lambda$ would likely be more appropriate.
>
> **[W3]** Our current formulation intentionally focuses on the standard RLVR setting with binary verifiable outcome rewards, where the Bernoulli likelihood and Beta posterior provide a clean and analytically tractable foundation for studying reward estimation under limited rollouts. We chose this setting to isolate the core contribution of the paper: reframing advantage computation as a reward-distribution estimation problem and showing that discounted Bayesian shrinkage can substantially reduce variance and improve MSE in the non-stationary on-policy regime. That said, we agree that the broader Bayesian perspective is not limited to Beta-Bernoulli rewards.
>
> More generally, the key idea is to maintain discounted sufficient statistics for a non-stationary reward distribution and use the resulting posterior mean and uncertainty for more stable advantage estimation. For bounded continuous rewards, one could replace the Beta-Bernoulli pair with an appropriate bounded likelihood model; for real-valued dense rewards, a Gaussian model with discounted conjugate updates would be a natural extension. Similarly, process-level rewards could be handled by applying the same discounted posterior estimation at the step- or segment-level, rather than only at the final outcome level. We will add a short discussion in the revision to clarify that DBB is one instance of a broader discounted Bayesian reward-estimation framework, while leaving the empirical study of continuous/dense process rewards to future work.

---

> > ### Author Rebuttal · Reviewer_pjp8 · 2026-04-01
> >
> > The authors have adequately addressed my main concerns raised in the original review. In particular, their rebuttal clarifies the technical points I was uncertain about and provides sufficient explanations regarding the experimental/theoretical issues. Based on these clarifications, my previous concerns are fully resolved.

---

> > > ### Author Response · Authors · 2026-04-04
> > >
> > > We are delighted that all of your concerns have been resolved. We sincerely appreciate your precise and insightful comments, which have helped strengthen the empirical aspects of our work.
> > >
> > > If your concerns have been fully resolved, may I kindly ask whether you would consider updating your score accordingly?
> > >
> > > Any additional questions or further discussion would be greatly appreciated and highly motivating for us. Please feel free to raise any further questions at any time.

---

### Official Review · Reviewer_HL9z · 2026-03-13

**Soundness:** 3
**Presentation:** 2
**Significance:** 3
**Originality:** 3
**Overall Recommendation:** 4
**Confidence:** 4

**Summary:**

This paper reframes reinforcement learning with verifiable rewards (RLVR) as a statistical estimation problem and proposes Discounted Beta–Bernoulli (DBB) reward estimation to improve sample efficiency in group-based methods such as GRPO. DBB maintains a per-prompt Beta posterior with exponential discounting to account for non-stationary on-policy training, providing a shrunk, lower-variance estimate of the Bernoulli reward probability that theoretically prevents variance collapse and empirically reduces MSE in low-sample regimes. Integrated into GRPO (and Dr.GRPO), DBB yields consistent gains across six in-distribution math reasoning benchmarks and three out-of-distribution datasets for Qwen3-1.7B and 8B models, reportedly with no extra rollout cost or memory.

**Compliance With Llm Reviewing Policy:**

Affirmed.

**Key Questions For Authors:**

- How exactly is the advantage computed under DBB for GRPO? Is μ set to $α/(α+β)$ and $σ^2$ set to p̂(1−p̂), or do you use the Beta–Bernoulli posterior predictive variance $(αβ)/[(α+β)^2(α+β+1)]$? Please clarify Eq. (13) and the normalization used in code.

- PPO clipping: You use (0.98, 0.98) for DBB vs (0.2, 0.28) for baselines. Can you provide controls where (a) GRPO uses the same wide clipping as DBB, and (b) DBB uses the baseline clipping, to isolate the estimator’s effect? Also report the fraction of updates clipped across methods.

- Could a simple per-prompt EMA of the empirical success rate (without Beta prior) or Laplace smoothing achieve similar gains? Please add these baselines or discuss why DBB outperforms them.

- How many random seeds were used, and what is the variance across runs, especially on OOD benchmarks? If single-seed, can you provide multi-seed results?

- Can you provide sensitivity to α0,β0 and discuss whether a hierarchical/shared prior (e.g., shrinkage to a global rate across prompts) could further improve robustness?

**Limitations:**

- The paper is missing several simple and relevant baselines, such as variance floors, Laplace smoothing, exponential moving averages of per-prompt success rates, or other lightweight shrinkage-based estimators. Without these, it is difficult to determine how much of the benefit is specific to DBB.

- The broader relation to classical shrinkage estimators and simple Bayesian smoothing methods is underdeveloped. A clearer comparison to EMA-style, Laplace, or hierarchical-shrinkage estimators would strengthen the empirical and conceptual positioning of the method.

- The exact uncertainty term used for normalization is not fully clear. The paper appears to use a variance based on the posterior mean rather than the standard Beta-Bernoulli posterior predictive variance, and this choice should either be justified more carefully or corrected.

- Some of the theoretical claims are stated too strongly. In particular, the improvement in mean-squared error over point estimation appears to hold only under certain conditions on the discount factor and non-stationarity, rather than universally.

**Strengths And Weaknesses:**

**Strengths**

- Recasts group-based RLVR advantage computation as distributional reward estimation, introducing a principled, lightweight Bayesian shrinkage scheme with exponential forgetting tailored to the non-stationary on-policy setting.
- Derives closed-form expressions for expectation, variance, and MSE of the DBB estimator under non-stationary reward probabilities, clarifying the bias–variance trade-off and conditions under which DBB improves over point estimation.
- Includes ablations on the discount factor λ and empirical MSE analyses as a function of λ and rollout count N, linking improved estimation to downstream performance.
- Clear conceptual framing (reward as a policy-induced distribution), with concise algorithm summary and intuitive explanation of shrinkage.
- Addresses core pain points (variance collapse, low-sample instability) in RLVR with a simple method that can be adopted widely and at scale.

**Weaknesses**

- The “posterior variance” used for normalization (Eq. 13) omits the standard Beta–Bernoulli predictive factor $(αβ/[(α+β)^2(α+β+1)])$; it appears to substitute Bernoulli variance at the posterior mean, which should be justified or corrected.
- The claim “MSE(DBB) < MSE(point)” is stated strongly in places, but the derivation only shows conditions and the empirical section shows it does not hold for all λ; phrasing should be tempered or accompanied by conditions.
- Missing strong baselines: comparison to simple and widely used fixes is absent (e.g., variance floors/epsilon in σ, Laplace/pseudocount smoothing without history, per-prompt EMA of success rates, dynamic-sampling/variance-aware variants like DAPO strategies, or GRESO). These could narrow or explain gains.
- No reporting of variance across random seeds; RLVR training outcomes can vary substantially, especially on OOD tasks, so statistical significance is unclear.
- A few inconsistencies/typos (e.g., “HBB” in Section 4, ambiguous clip ranges and “clip-higher” terminology) and notational slippage (epoch vs iteration index) detract from clarity.
- The precise advantage computation under DBB for GRPO (is σ from posterior predictive variance or Bernoulli variance at the posterior mean?) should be stated explicitly.

---

> ### Author Rebuttal · Authors · 2026-03-31
>
> We thank the reviewer for their thoughtful feedback and recognition of our work’s practical relevance. We respond to their comments below. Figures and tables are given in **[LINK](https://github.com/anonyauthor1872/2026-icml-rebuttal)**.
>
> **[W1+W6+Q1]** As the advantage is calculated based on the mean and variance of rewards (Equation (2)), we use those of the Bernoulli reward distribution, which are shown in Equations (12) and (13). We note that this should not be confused with $\frac{\alpha \beta}{(\alpha + \beta)^2 (\alpha + \beta + 1)}$, which corresponds to the variance of the probability $p \sim \text{Beta}(\alpha, \beta)$.
>
> **[W2]** We agree with the reviewer and will revise the claim to be more precise by explicitly stating the conditions under which it holds.
>
> **[W3+Q3]** We first formulate the estimators and their MSE for the per-prompt exponential moving average (EMA) and Laplace smoothing (Lap) as follows:
>
> $$
> \hat{p}^{\text{EMA}}_\tau = (1-\lambda)\,\frac{S_\tau}{N} + \lambda\,\hat{p}^{\text{EMA}}_{\tau-1}
> \quad \text{and} \quad
> \mathrm{MSE}(\hat{p}^{\text{EMA}}_\tau) = \left(\sum_{k=1}^{\tau-1} w_k (p_k - p_\tau)\right)^2 + \sum_{k=1}^{\tau} w_k^2 \cdot \frac{p_k(1-p_k)}{N}
> $$
>
> $$
> \text{where} \qquad
> w_1 = \lambda^{\tau-1}, \quad
> w_k = (1-\lambda)\lambda^{\tau-k} \;\; (k \geq 2)
> $$
>
> $$
> \hat{p}^{\text{Lap}}_\tau
> = \frac{S_\tau + \lambda}{N + 2\lambda}
> \quad \text{and} \quad
> \mathrm{MSE}(\hat{p}^{\text{Lap}}_\tau)
> = \frac{\lambda^2 + (N - 4\lambda^2)\, p_\tau(1-p_\tau)}{(N+2\lambda)^2}
> $$
>
> Based on these formulations, we conduct an MSE comparison experiment under the same setting as in Section 5.3.
>
> As shown in Figure 6, GRPO-DBB achieves the lowest MSE of 0.0083 at $\lambda=0.4$, outperforming all other methods. EMA attains its best performance at $\lambda=0.3$ with an MSE of 0.0086, which is 2.67\% higher than that of GRPO-DBB. However, as $\lambda$ increases, the MSE of EMA rises rapidly, indicating reduced stability. Moreover, the range of $\lambda$ values for which EMA outperforms GRPO is relatively narrow, suggesting high sensitivity to hyperparameter selection. Similarly, Laplace smoothing achieves its lowest MSE of 0.0108 at $\lambda=0.3$, which is 21.71\% higher than that of GRPO-DBB. Like EMA, it only outperforms GRPO within a limited range of $\lambda$, again indicating strong sensitivity to the choice of hyperparameter.
>
> Therefore, DBB can be interpreted as the most favorable option, as it achieves the lowest MSE at its optimum and maintains a broader range of $\lambda$ for which it outperforms GRPO. For other baselines, please refer to the response to the reviewer pjp8's W1.
>
> **[W4+Q4]** Please refer to the response to the reviewer pjp8's W4.
>
> **[W5+Q2]** We will correct typos and inconsistencies in the revision. For details on clipping, please refer to the response to the reviewer TVSP's W1.
>
> **[Q5]** We measure MSE with respect to $\alpha_0,\beta_0$, and $\lambda$ under the same setup as in Section 5.3. Figure 7(a) shows that MSE can vary depending on the choice of $(\alpha_0,\beta_0)$, particularly when these parameters take large values. As $\alpha_0+\beta_0$ increases, the influence of the $N$ on-policy rewards diminishes. Consequently, the estimator becomes more sensitive to $\lambda$, which can amplify bias and lead to higher MSE (Figure 7(b)).
>
> In practice, the Beta distribution is typically initialized with $(\alpha_0,\beta_0)=(1,1)$. This is because, in the absence of prior knowledge about the underlying Bernoulli distribution, setting asymmetric initial values may introduce bias. This symmetric choice is both reasonable and robust, as it achieves relatively low MSE across a wide range of $\lambda$ values while maintaining a competitive minimum MSE.
>
> Overall, although the MSE is sensitive to $(\alpha_0,\beta_0)$ and the corresponding $\lambda$, this sensitivity does not pose a practical concern as long as pathological choices are avoided.
>
> We also consider Empirical Bayes (EB) as a representative approach for introducing a hierarchical/shared prior. The hyperparameters of the shared Beta prior across prompts, $\alpha_{EB}$ and $\beta_{EB}$, are estimated via online method-of-moments, resulting in $\hat\alpha_{EB}$ and $\hat\beta_{EB}$.  Under this formulation, the estimator in online step $s$ can be written as:
>
> $$\hat{p}_\tau^{EB} = \frac{N\,\hat{p}_\tau^{pt} + \hat\alpha_{EB}(s)}{N + \hat\alpha_{EB}(s) + \hat\beta_{EB}(s)}$$
>
> Based on this estimator, the MSE can be expressed as:
>
> $$\text{MSE}(\hat{p}_\tau^{EB}) = \frac{\nu_{EB}^2\left(m_{EB} - p_\tau \right)^2 + N\,p_\tau(1-p_\tau)}{(N + \nu_{EB})^2},$$
>
> where
>
> $\nu_{EB}(s) = \hat\alpha_{EB}(s) + \hat\beta_{EB}(s)$
>
> and
>
> $m_{EB}(s) = \hat\alpha_{EB}(s)/\nu_{EB}(s)$.
>
> EB has the advantage of not requiring manual hyperparameter tuning, but its minimum achievable MSE is higher than that of GRPO-DBB (Figure 8). This suggests a trade-off between robustness (i.e., reduced dependence on $\lambda$) and optimal performance.

---

> > ### Author Rebuttal · Reviewer_HL9z · 2026-04-04
> >
> > Thank you for the rebuttal. Most of my concerns have been resolved, and I appreciate the authors’ thoughtful responses and effort in addressing them. I encourage the authors to incorporate these clarifications into the revised version.

---

> > > ### Author Response · Authors · 2026-04-04
> > >
> > > We would also like to sincerely thank you for your thoughtful and insightful review. We believe your feedback has helped strengthen both the empirical and theoretical justification of our work.
> > >
> > > We will incorporate the additional experimental results obtained during the rebuttal process into the revised version of the paper.
> > >
> > > If your concerns have been fully addressed, may I kindly ask whether you would consider updating your score accordingly.
> > >
> > > We welcome any further questions or discussion. Thank you.

---

### Official Review · Reviewer_TVSP · 2026-03-13

**Soundness:** 3
**Presentation:** 3
**Significance:** 2
**Originality:** 3
**Overall Recommendation:** 5
**Confidence:** 3

**Summary:**

The paper proposes using a discounted Beta-Bernoulli distribution to track historical rewards, replacing the standard batch mean and standard deviation computations in GRPO advantage estimation. This approach aims to prevent variance collapse, a situation where all rollouts in a group receive identical rewards.

**Compliance With Llm Reviewing Policy:**

Affirmed.

**Final Justification:**

The author has fully addressed my main concerns.

**Key Questions For Authors:**

Q1. In Appendix C, an important observation is made without theoretical explanation: "For RLVR-DBB, the importance ratios tend to be larger than those of naive methods". This leads to the highly questionable decision to set the clipping range for RLVR-DBB to $(0.98, 0.98)$, which is massively wider than the $(0.2, 0.28)$ range used for GRPO and RePO. Mechanically, these larger importance ratios likely occur because DBB abandons GRPO's zero-sum advantage normalization, leading to unconstrained update sizes. A wider clip range inherently allows for larger gradient magnitudes and faster learning. To isolate the true effect of RLVR-DBB from simply taking larger gradient steps, please report the baseline results trained under this exact same $(0.98, 0.98)$ clipping range.

Q2. Figure 2 shows a clear trend for both model scales: the entropy of GRPO-DBB consistently decreases over the course of training, whereas standard GRPO's entropy exhibits a U-shape. How does replacing the group-relative mean with a historical moving average induce this specific, continuous drop in entropy? Additionally, please explain why the response length for the 1.7B model increases so significantly. Finally, please provide a plot showing Best@8 and/or Mean@8. Usually, lower entropy corresponds to a higher Mean@8, but Best@8 may suffer as exploration decreases.

Q3. The paper claims that GRESO cannot fully solve the variance collapse problem of GRPO. Since RLVR-DBB is presented as a solution to this exact problem, can you demonstrate whether RLVR-DBB empirically outperforms GRESO in terms of downstream performance versus rollout utilization?

**Limitations:**

The authors did not discuss limitations

**Strengths And Weaknesses:**

Strength:

- The paper provides a principled way to address the variance collapse problem inherent in standard GRPO.
- The writing is generally clear and logically structured.
- The proposed algorithmic fix is theoretically sound and relatively simple to implement.

Weakness:

- W1. The experimental evidence is weakened by a significant hyperparameter mismatch between the proposed method and the baselines (see Q1), as well as insufficient explanations for certain training dynamics (see Q2).

- W2. While framed as a principled solution, the method relies heavily on a sensitive, empirically tuned hyperparameter discount factor $\lambda$. The fact that it requires a drastically different value depending on the model scale ($\lambda=0.5$ for 1.7B and $\lambda=0.75$ for 8B) introduces a heuristic vulnerability

- W3. The paper would benefit from evaluating against more of the specific baselines discussed in its related work section, such as GRESO.

- W4. There is insufficient explanation for some of the observed training dynamics, particularly the diverging entropy curves and response lengths (see Q2).

---

> ### Author Rebuttal · Authors · 2026-03-31
>
> We thank the reviewer for their insightful and constructive feedback. We address their comments below.
> Figures and tables are given in **[LINK](https://github.com/anonyauthor1872/2026-icml-rebuttal)**.
>
> **[W1+Q1]** As you mentioned, the larger importance ratios in GRPO-DBB stem from removing zero-sum advantage normalization. Specifically, GRPO-DBB yields smaller reward variance (except in the zero-variance case), leading to larger advantage values, which in turn produce larger gradient norms and importance ratios (Figure 4(c) and (d)). Thus, these larger updates are inherent to the DBB estimator, not a result of using a wider clipping range. Importantly, a wider clipping range does not necessarily increase gradient magnitude or improve performance, as GRPO shows similar or even smaller gradient norms and no clear policy gradient loss improvement under a wider range.
>
> Under the standard range (0.2, 0.28), GRPO-DBB experiences 2-3 times higher clipping rates than GRPO (Figure 4(b)). This excessive clipping suppresses a substantial portion of meaningful training signals, which degrades the performance (Figure 4(a)). However, under a wider clipping range (0.98, 0.98), GRPO-DBB outperforms GRPO, indicating that the performance gains are not solely due to taking larger update steps. Accordingly, the final results are reported with the best configuration for each method.
>
> **[W2]** We agree that searching for the optimal $\lambda$ can be computationally expensive. However, we would like to emphasize that GRPO-DBB consistently outperforms GRPO across almost all $\lambda$ values, except for only the case of 1.0 in the 1.7B model. Therefore, while selecting the optimal $\lambda$ may involve some heuristics, our method robustly outperforms GRPO across a wide range of $\lambda$.
>
> To further reduce the heuristic burden in choosing $\lambda$, we provide an empirical guideline. Specifically, we conduct an additional $\lambda$ search experiment on a smaller model, Qwen3-0.6B-Base, which achieves the best performance at $\lambda=0.25$ (Table 7). Combined with our earlier observations that optimal $\lambda$ is 0.5 and 0.75 for the 1.7B and 8B models, respectively, this suggests a consistent trend: smaller models tend to benefit from smaller $\lambda$.
>
> More generally, this can be interpreted by the bias-variance trade-off of the DBB estimator. As $\lambda$ increases, variance decreases, but bias may increase, particularly when the reward distribution shifts significantly across training epochs. With weaker models or challenging datasets, such shifts tend to be larger, making a smaller $\lambda$ preferable. In contrast, when the model performs well on the train dataset, the reward distribution has a natural upper bound, making a larger $\lambda$ beneficial by reducing variance more than it increases bias, thus lowering overall MSE.
>
> **[W3+Q3]** Please refer to the response to the reviewer pjp8's W1.
>
> **[W4+Q2]** A decrease in entropy indicates that some token trajectories become less probable and are pruned from the policy model's token tree. Compared to GRPO, this effect is particularly pronounced in GRPO-DBB for cases where all rewards within a group are zero. In such cases, GRPO has no training signal, while GRPO-DBB produces negative advantages, rapidly suppressing incorrect trajectories, inducing a reduction in entropy. We interpret this behavior as a key mechanism underlying the performance gains of GRPO-DBB over GRPO.
>
> The increase in response length follows from this entropy dynamic. To obtain a reward, the model can explore (i) breadth-wise (diversifying trajectories) or (ii) depth-wise (extending the length of trajectories). Under GRPO-DBB, stronger pruning makes the trajectory distribution effectively shallower than that of GRPO, limiting breadth-wise exploration. Consequently, the model relies more on depth-wise exploration, producing significantly longer responses than GRPO.
>
> Figure 5 illustrates Mean@8 and Best@8 on the validation set during training. Note that the Acc@8 reported in the paper is equivalent to Mean@8. For the 1.7B model, GRPO-DBB consistently achieves higher Best@8 than GRPO throughout most of training. For the 8B model, although GRPO temporarily attains higher Best@8 during the middle stages of training, GRPO-DBB ultimately surpasses GRPO in the best performance. These results indicate that, despite reduced entropy, GRPO-DBB does not compromise and, in fact, improves Best@8 in the long run.
>
> To assess whether the Best@8 improvements of GRPO-DBB are statistically significant, we further evaluated the best checkpoint of each method on six in-distribution evaluation sets using four independent evaluation runs (i.e., 32 responses per prompt). GRPO achieves Best@8 scores of 61.75±0.92 (8B) and 39.79±0.26 (1.7B), while GRPO-DBB achieves notably stronger performance, reaching 63.61±1.16 and 43.73±0.43, respectively.

---

> > ### Author Rebuttal · Reviewer_TVSP · 2026-04-03
> >
> > I thank the authors for their thorough rebuttal. The additional ablations and explanations have addressed my primary concerns. I will raise my score from a 3 to a 5.
> >
> > **Additional Questions:**
> > 1. How does the DBB framework operate under non-binary reward signals? For example, it is common for RLVR methods to apply length penalties, resulting in continuous training signals rather than strictly binary ones.
> > 2. Would it be feasible and beneficial to learn a prompt-specific $\lambda$ to account for varying learning difficulties, rather than relying on a global hyperparameter?
> >
> > **Additional Comment:**
> > Variance collapse or vanishing advantages are highly active research topics surrounding the GRPO algorithm. The paper would benefit from a broader discussion of alternative methods for handling variance collapse to better contextualize the contribution. For example, [1] and [2] rely on dynamic rollout allocation to minimize wasting rollouts on zero-variance prompts. Additionally, [3] and [4] focus on improving advantage diversity to reduce the zero-advantage problem.
> >
> > **References:**
> >
> > [1] Yixuan et al., Not All Rollouts are Useful: Down-Sampling Rollouts in LLM Reinforcement Learning
> >
> > [2] Nguyen et al., Adaptive Rollout Allocation for Online Reinforcement Learning with Verifiable Rewards
> >
> > [3] Xingjian et al., EDGE-GRPO: Entropy-Driven GRPO with Guided Error Correction for Advantage Diversity
> >
> > [4] Wang et al., VL-Rethinker: Incentivizing Self-Reflection of Vision-Language Models with Reinforcement Learning

---

> > > ### Author Response · Authors · 2026-04-04
> > >
> > > We are glad that the reviewer’s concerns have been resolved, and we sincerely appreciate your decision to raise the score.
> > > We are also grateful for your sharp and thoughtful review, which has helped strengthen our work both theoretically and empirically.
> > >
> > > We provide responses to your additional questions below.
> > >
> > > **1. Continuous reward settings in the DBB framework.**
> > > Extending the DBB framework to continuous reward settings is indeed an important direction that we, as authors, are actively pursuing as future work. Our response to Reviewer pjp8’s W3 also outlines related plans and directions.
> > >
> > > In brief, when continuous rewards (e.g., length penalties) are considered, the Beta-Bernoulli model can be naturally replaced by a Gaussian process formulation. Since Gaussian process is also conjugate of itself, the core idea of DBB—leveraging discounted historical observations for distributional estimation—can be utilized in this setting.
> > >
> > > Through DBB, we empirically observe that accurately estimating the reward distribution using historical rewards has a direct impact on the performance of RLVR algorithms. While we do not yet have experimental evidence in continuous settings, we hypothesize that reducing the MSE of reward distribution estimation via Gaussian processes would similarly lead to performance improvements in RLVR.
> > >
> > > **2. Prompt-specific discount factors.**
> > > We agree that this is a promising direction. More precisely, since the learning difficulty of each prompt (i.e., the degree of shift in the reward distribution across epochs) varies across prompts, using prompt-specific $\lambda$ values could improve the accuracy of reward distribution estimation.
> > >
> > > A practical approach would be to start with a global $\lambda$ during the first epoch. From the second epoch onward, one can estimate how much the reward distribution has shifted for each prompt and adapt $\lambda$ accordingly. This would allow for more fine-grained and accurate estimation, which we expect to have a positive impact on overall performance.
> > >
> > > We hope our responses are helpful. Once again, we sincerely thank you for your thoughtful feedback, which has significantly improved our work.

---

### Decision · Program_Chairs · 2026-04-30

**Decision:**

Accept (regular)

**Comment:**

There is general agreement that the paper provides a principled solution to the problem of variance collapse in GRPO by appealing to Bayesian conjugate modeling. The rewards are binary and are treated as a `Bernoulli distribution and the prior as a Beta distribution. Since the trajectory is non-stationary, they use simple discounted moving average to update the posteriors and that works. There obtain substantial gains on MMLU-Pro which attests to handing OOD data. Majority of the concerns of reviewers were addressed to their satisfaction.